# Boosting Large Language Models with Mask Fine-Tuning

## Abstract

The large language model (LLM) is usually kept integral in the mainstream optimization protocol. No work has questioned whether maintaining the model integrity is *indispensable* for promising performance. In this work, we introduce Mask Fine-Tuning (MFT), a brand-new LLM fine-tuning paradigm to show that properly breaking the model's structural integrity can surprisingly lead to improved performance without model weight updates. Specifically, MFT learns and applies a set of binary masks on well-optimized models supervised by the typical LLM fine-tuning objective. Based on fully fine-tuned models, MFT uses the same fine-tuning datasets to gain consistent performance boosts across various domains and backbones (e.g., **2.60 / 4.15** average gain in IFEval with LLaMA2-7B / 3.1-8B). Detailed ablations and analyses study the proposed MFT from different perspectives, such as sparse ratio, loss surface, etc. Additionally, by deploying it on well-trained models, MFT is compatible with collaborating with other LLM optimization procedures for general model enhancement. Further, this study extends the functionality of the masking operation from its conventional network pruning context for model compression into a general model capability scope.

## 1 Introduction

Pre-training large language models (LLMs) typically requires massive corpora to enable models with foundational knowledge and linguistic competencies for effective language generation (Radford et al., 2019; Brown et al., 2020; Touvron et al., 2023; Grattafiori et al., 2024). After this, the pre-trained LLMs are transferred for downstream tasks by specific fine-tuning using high-quality domain data, which includes specialized knowledge such as math and coding (Cobbe et al., 2021; Yu et al., 2024; Chen et al., 2021; Austin et al., 2021), or particular patterns such as human instruction (Zhou et al., 2023; Dubois et al., 2024). For such adaptations, fine-tuning all model parameters is the most straightforward and common method, commonly referred to as full fine-tuning (FFT), which generally achieves the most competitive performance. Furthermore, adapter-based alternatives offer parameter-efficient fine-tuning (PEFT) strategies such as vanilla bottleneck adapters (Houlsby et al., 2019) and LoRA (Hu et al., 2022). These methods fix the pre-trained backbone and additionally introduce a limited number of learnable parameters. Even if such methods may sacrifice fine-tuning performance compared with FFT, they are well suited to scenarios with limited computational budget, especially for insufficient fine-tuning data. Overall, the pipeline of language model pre-training followed by fine-tuning has demonstrated promising results across a range of language tasks, including domains requiring extensive specialized knowledge (e.g., medicine (Wang et al., 2023) and law (Nguyen, 2023)) and complex reasoning (e.g., math (Frieder et al., 2024) and coding (Chen et al., 2021)), exhibiting remarkably intelligent capabilities.

However, such mainstream LLM optimization process (pre-training followed by fine-tuning), promisingly benchmarking modern LLM performances, constantly treats the language model structure as a whole. For pre-training, the model structure is mostly kept dense. For fine-tuning, FFT optimizes all model parameters simultaneously, and PEFT freezes it to tune additional adapter parameters, where both accept the necessity of LLM structural integrity by default. Such practice naturally inspires us to ask: *Is such structural integrity indispensable for good model performance?*, or even further, is there potential to further improve the model by removing certain model components that break the structural integrity?

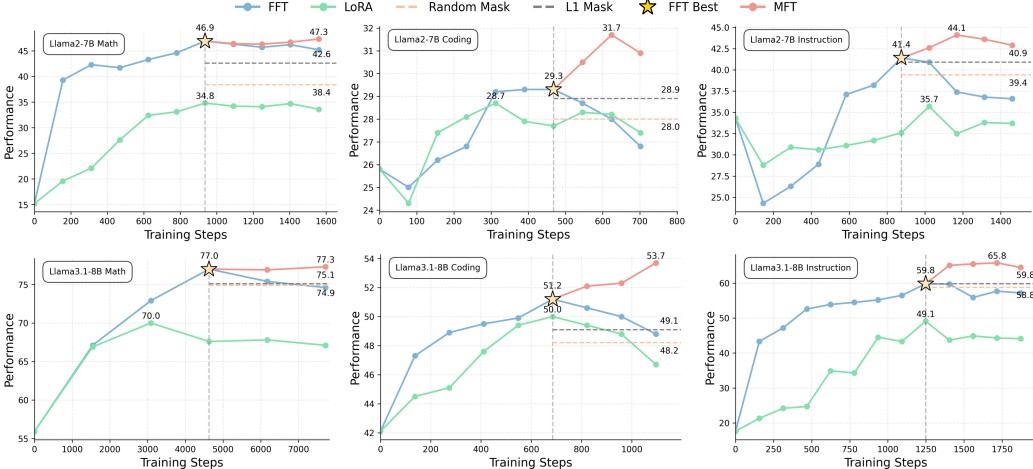

Figure 1: The visualization of performance trend along with different fine-tuning strategies, including FFT (blue line), LoRA (green line) (Hu et al., 2022), and our MFT (red line). We also add random (orange dash) and L1 masks (gray dash) for comparison. We use three settings across LLaMA2 and LLaMA3.1 backbones on GSM8K, HumanEval, and IF-Eval for math, coding, and instruction domains, respectively. The x-axis is training steps starting from the pre-trained backbone. The y-axis is evaluation performance. MFT (red line) starts from the best FFT model (yellow star) and breaks the upper bound with further improvements, while continued FFT leads to over-fitting. It also performs better than LoRA fine-tuning and two vanilla mask baselines.

We propose *mask fine-tuning (MFT)* to answer this question. MFT freezes a given LLM model and learns a binary mask on it to check if breaking structural integrity works. To substantially validate its effectiveness, we choose a well-trained LLM (model after sufficient FFT in this study, serving as a representative strong baseline) as a starting point to see if MFT makes further improvement. MFT uses the regular LLM fine-tuning objective and datasets to learn the mask, which are the same as the ones for the FFT. The only difference is that the learnable part is changed from model weights to a binary mask of these weights, while the given weights are fixed. The learned mask indicates positions of certain parameters to be removed. Surprisingly, we find masking these weights leads to further performance gain compared with the given well-trained model. As shown in Fig. 1, regular FFT (blue line) improves the pre-trained model for different settings, while after reaching the best performance (yellow star), excessive FFT causes overfitting. However, MFT (red line) starts from the best FFT model and further improves it. These teaser cases show a consistent observation across different backbones (LLaMA2-7B (Touvron et al., 2023) and LLaMA3.1-8B (Grattafiori et al., 2024)) and domains (GSM8K (Cobbe et al., 2021), HumanEval (Chen et al., 2021), and IF-Eval (Zhou et al., 2023) for math, coding, and instruction, respectively). Comparisons of more configurations are elaborated in Sec. 3. Such a phenomenon substantially answers the question we are interested in: *the LLM structural integrity is not indispensable for good model performance, and breaking such integrity can lead to further improvements*.

Herein, we mainly explore MFT as a post fine-tuning strategy with further improvement, particularly starting from a yet outperforming the best FFT checkpoint. It naturally upgrades the current fine-tuning routine as a new protocol shown in Fig. 2, providing a new perspective to investigate LLM fine-tuning. Further, MFT is compatible with other existing fine-tuning methods and can be integrated into any of them flexibly, as it follows the same optimization objective and requires no additional data annotation.

Besides, we emphasize the difference between typical pruning methods and our work. The former one compresses the model and tries to maintain the trained model's capability, but the latter one aims for further improvement above a well-trained model without the purpose. Even if they share the same concept of *mask* (or model sparsity), their fundamental goals are different (see more discussions in Sec. 4). Following this line, our study extends the functionality of the masking operation from typical network compression into a more general model capability scenario. In other words, analogically speaking, *typical masking employs subtraction to reduce (pruning to compress), whereas our approach leverages subtraction to achieve augmentation (removing weights to improve)*. We summarize our contributions as follows:

- We validate that a well-trained LLM can be further improved by carefully removing certain weights using the same objective and dataset as regular fine-tuning, with limited computational overhead. Such a strategy is compatible with working with other training pipelines and upgrades the current LLM optimization protocol.

- We propose *Mask Fine-Tuning (MFT)* to complete our exploration in a post fine-tuning scenario. MFT starts from a competitive, fully fine-tuned model with fixed weights and learns a binary mask applied to the given model to improve it. Following this, MFT treats model sparsity in a new light, not only for efficiency, but also for performance improvement.

- Extensive experiments across different backbones (LLaMA2 and LLaMA3.1), domains (math, coding, and instruction), and FFT settings (domain-specific and mixed-up) show the effectiveness of MFT with consistent performance gain. Detailed ablations and analyses are provided for a better intuition to inspire more future works.

## 2 MASK FINE-TUNING

We focus on GPT-like language models (Brown et al., 2020; Radford et al., 2019) in an auto-regressive fashion and start from a fully fine-tuned model of a pre-trained language backbone to conduct our mask fine-tuning (MFT). Therefore, we briefly introduce the language model notations of full fine-tuning (FFT), then introduce our MFT.

**Full Fine-Tuning.** We refer a pre-trained auto-regressive language backbone as $\mathcal{N}_p$ with parameters $\Theta_p$ and optimize the following objective to represent a FFT:

$$L(U_f) = \sum_i logP(u_f^i|u_f^{i-k}, ..., u_f^{i-1}; \Theta_p), \quad (1)$$

Figure 2: Typical LLM training contains pre-training and fine-tuning for foundation capacity and domain knowledge, where the structure is always entire. We are curious if such integrity is necessary and propose MFT to generally outperform models with sufficient FFT, naturally upgrading the classic pipeline following the typical protocol to further refine well-optimized LLMs.

where $L$ represents the auto-regressive loss to supervise the next token prediction. $\Theta_p$ is fully optimized, and $U_f = \{u_f^1, ..., u_f^n\}$ is a token sequence of a language corpus, serving as an FFT dataset. We refer to the model after sufficient FFT as $\mathcal{N}_f$ with optimized parameters $\Theta_f$.

**Mask Fine-Tuning.** We deploy our MFT on models after the FFT. Formally, we revise Eq. equation 1 by adding a binary mask onto model parameters and obtain the rewritten loss for MFT as below:

$$L(U_m) = \sum_i logP(u_m^i|u_m^{i-k}, ..., u_m^{i-1}; \Theta_f \odot M), \quad (2)$$

where it shares the same objective function $L$ as FFT with the dataset $U_m$ for supervision. Herein, the dataset $U_m$ for MFT is the same as $U_f$ for FFT in our experiments. The main difference is that we add a binary mask $M$ on the given model parameters $\Theta_f$, where $M$ and $\Theta_f$ share the same size and corresponding weights on $\Theta_f$ are masked out by element-wise multiplication $\odot$. During MFT, $\Theta_f$ is fixed and $M$ is updated. This way, MFT learns to find locations of certain weights while keeping the fully fine-tuned parameters $\Theta_f$ unchanged. And these weights can be removed from the well-trained model by applying this learned binary mask. $M$ is optimized by a learnable process leveraged on the straight-through gradient estimator (Bengio et al., 2013), and we name it the learnable mask.

**Learnable Mask.** Given a model parameter $\Theta_f$ after FFT, we represent it in a layer-wise fashion as $\Theta = \{\theta_l\}, l \in \{1, 2, ..., L\}$, where we omit the subscript $f$ for notation simplicity. $L$ represents the model layer number. Then, we can formalize each layer as

$$I_{l+1} = \sigma(F[I_l; \theta_l]), \quad (3)$$

where $I_l$ is the layer $l$ input and $I_{l+1}$ is its output with activation $\sigma$. $F$ generally serves as a layer operation (e.g., a convolutional or linear layer) with a parameter $\theta_l = \{\theta_l^d\}, d \in \{1, 2, ..., D_l\}$. $D_l$ represents the layer $l$ parameter dimension. In this work, $F$ represents a linear layer, as we mainly consider Transformer-based language models and deploy the learnable mask on model linear mappings. To perform mask learning, we fix all parameters $\Theta$, denoted as $\overline{\Theta}$. Then, each weight $\overline{\theta_l^d}$ is assigned with a learnable score $c_l^d$, representing the importance of corresponding weight. Based on this, we rewrite Eq. equation 3 as

$$I_{l+1} = \sigma(F[I_l; \overline{\theta}_l \odot v(c_l)]), \tag{4}$$

where $c_l = \{c_l^1, c_l^2, ..., c_l^{D_l}\}$ is the score of parameter $\overline{\theta}_l$. $v$ is an indicator function to select a mask based on scores. In this work, we use a ratio-based version, where $v$ outputs 1 or 0 depending on whether the value of $c_l^d$ is in the top $K\%$ highest scores or not, and $K$ is a pre-defined mask ratio. By updating $c$ with fixed $\overline{\Theta}$, part of the parameters in $\overline{\Theta}$ are kept while others are masked out. As the indicator function $v$ is non-differentiable, we use the straight-through gradient estimator (Bengio et al., 2013) to estimate the gradient of the loss with respect to $c_l^d$. Concretely, $v$ is regarded as an identity function during the gradient backwards step, then the approximation can be described as

$$\tilde{g}(c_l^d) = \frac{\partial L}{\partial \tilde{I}_{l+1}} \frac{\partial \tilde{I}_{l+1}}{\partial c_l^d} \approx \frac{\partial L}{\partial I_{l+1}} \frac{\partial I_{l+1}}{\partial c_l^d}, \tag{5}$$

where $\tilde{I}_{l+1} = \sigma(F[I_l; \overline{\theta}_l \odot c_l])$ after applying estimation and $\tilde{g}(c_l^d)$ is the estimated gradient with respect to score $c_l^d$. In this way, MFT learns to obtain a mask for model $\mathcal{N}_f$, and we denote the final model as $\mathcal{N}_m$ with parameter $\Theta_f$ and mask $M$.

MFT learns to identify certain weights guided by the regular training loss of language models, and further removes them by applying the learned mask. It answers our question (Sec. 1) and provides a new LLM fine-tuning protocol. We illustrate the MFT procedure in Fig. 2. Please note, the FFT method used in this study is supervised fine-tuning (SFT) without considering other policy-based tuning like DPO (Direct Preference Optimization) (Rafailov et al., 2024) and PPO (Proximal Policy Optimization) (Schulman et al., 2017). Our MFT can be easily generalized to them, and we leave such explorations to our future work.

## 3 EXPERIMENTS

We evaluate the proposed mask fine-tuning (MFT) on large language models (LLMs) across several backbones, domains, and full fine-tuning (FFT) settings. In addition, we provide ablations and visualization analysis of different aspects to provide a better intuition.

### 3.1 EXPERIMENTAL SETUPS

**Pretrained Backbones.** We use Transformer (Vaswani, 2017) based pretrained LLMs as backbone models, including LLaMA2-7B (Touvron et al., 2023) and LLaMA3.1-8B (Grattafiori et al., 2024).

**Domains & Datasets.** We involve three domains with their representative tasks. Specifically, we include GSM8K (Cobbe et al., 2021) and MetaMath (Yu et al., 2024) for math, HumanEval and HumanEval+ (Chen et al., 2021) for coding, and IF-Eval (Zhou et al., 2023) and Alpaca-Eval (Dubois et al., 2024) for instruction following domains, respectively. All three of these domains are used for typical FFT, our MFT, and evaluation.

**Experimental Configurations.** Based on pretrained backbones, our experiments contain FFT, MFT, and downstream evaluation in order. For FFT, we follow two strategies: 1) domain-specific, 2) mix-up, where the former one conducts FFT using datasets from individual domain from math, coding, or instruction following, and the latter one using a mix-up dataset from all these domains. For MFT, we follow a domain-specific fashion, using datasets from one individual domain. More dataset details are provided in the supplementary. We mainly investigate local masking in this paper, which is elaborated by a series of proof-of-concept studies in Sec. 3.2 and Fig. 3. We also provide an initial exploration of global masking strategy in Sec. 3.3 and Tab. 3. For evaluation, we comprehensively test performance changes within (Tab. 1 and Tab. 2).

**Baselines.** We use several baselines for comparisons, including (1) a very basic pre-trained backbone; (2) FFT as a strong baseline, which is mostly seen as the most competitive approach for downstream fine-tuning. We perform a sufficient FFT and report the best performance; (3) LoRA (Hu et al., 2022) fine-tuning as a strong alternative of FFT; (4) continued FFT, which always causes a performance drop due to overfitting; (5) continued LoRA plays a similar role for the LoRA baseline. We conduct 4 epochs of training for all baselines to make them sufficient and report the final performances for Continued FFT and Continued LoRA. (6) random mask, and (7) L1 mask as two vanilla masking strategies. Among them, (2) FFT and (4) continued FFT are critical to our experiments, since our goal is to use MFT to further improve the best FFT, and continued FFT serves as a sanity check.

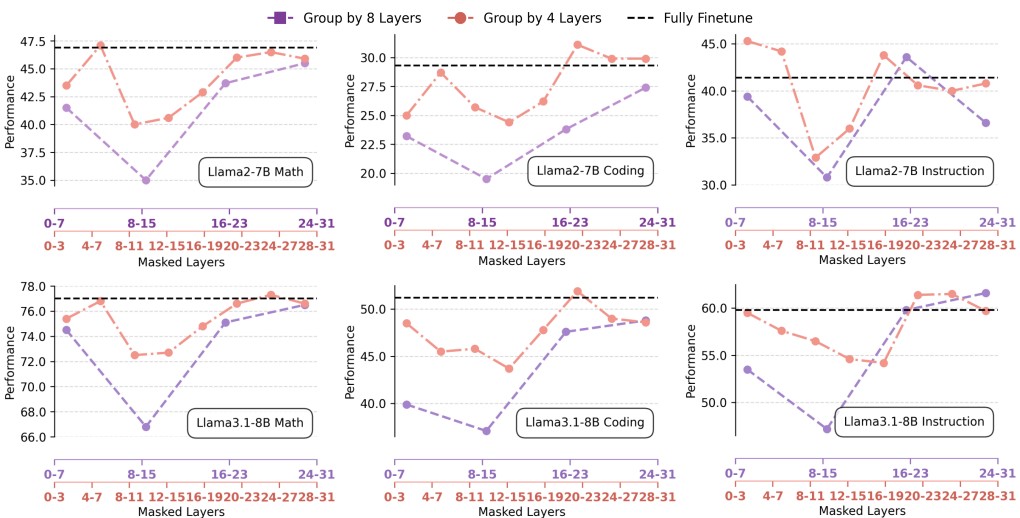

Figure 3: Visualization of ablation study of local MFT strategy. It uses LLaMA2-7B and LLaMA3.1-8B backbones, covering math, coding, and instruction domains. In each figure, we conduct MFT ablations with a 10% masking ratio on a domain-specific FFT model (black dashed line). We swap the ablation under two local granularities, 8-layer (purple) and 4-layer (orange), from shallow to deep layers with a 10% fine-tuning set. We find 1) MFT can outperform the FFT strong baseline and 2) MFT performs better in shallow (0-7) and relatively deep layers (20-27). This ablation is only for quick intuitions with limited performance gain of MFT using a 10% subset. Based on this trend, we deploy MFT with complete fine-tuning sets, achieving more improvements (Tab. 1 and Tab. 2).

## 3.2 MAIN RESULTS

We describe the MFT validation logic and draw conclusions. As mentioned in Sec. 3.1, MFT mainly uses a local masking strategy, therefore, we start from a proof-of-concept study to test if MFT has potential for further improvements (Fig. 3). Such ablation also helps us detect the most effective components of the well-trained model to employ MFT. Accordingly, we extensively deploy MFT in this local fashion across different configurations and compare with strong fine-tuning baselines.

**Proof-of-Concept Study of Local Mask Fine-Tuning.** This series of ablations aims to check if the MFT has the potential to improve model capability after FFT. If so, we expect a comprehensive ablation to also detect the sensitivity of different model components to MFT. Concretely, we start from LLaMA2-7B and LLaMA3.1-8B backbones after domain-specific FFT and conduct MFT using a small subset of each domain. Specifically, for each domain, we use 10% of its corresponding fine-tuning set. To swap the model, we set 4 or 8 layers in a row as a group to ablate the model from shallow to deep layers. For each given group, we use MFT to locally learn a mask and keep other layers fixed to check model performance. For example, for the layer 4 to 7, it is formally achieved by setting $\Theta = \{\theta_l\}, l \in \{4, 5, 6, 7\}$. Within each group, we set the masking ratio as 10% to learn the mask, which means we aim to learn and remove 10% parameters while keeping the remaining 90%.

Fig. 3 shows the detailed ablation results. We find that 1) compared with FFT (black dashed line), the MFT has the potential to further improve for each setting, even if they may have different proper layer-wise groups for MFT; 2) A smaller layer-wise group (4-layer) is better for MFT compared with a larger one (8-layer), however, they generally share a similar trend, where sçhallow (0-7) and mid-to-rear (20-27) partitions often respond positively to MFT than other parts. Such observation confirms the MFT's potential to further improve models after FFT, and then indicates promising partitions to conduct mask learning. Concretely, for LLaMA2-7B, we use 4-7, 20-23, and 0-3 for math, coding, and instruction domains, respectively, for MFT. For LLaMA3.1-8B, we use 24-27, 20-23, and 24-27 for these three domains accordingly. For all full dataset MFT results in the rest of this paper, we follow this proof-of-concept study to use a constant 10% sparse ratio.

Notably, 1) Since this ablation is only for quick intuition and sanity check of MFT effectiveness, we only use a 10% dataset subset here. According to such insights, MFT is trained using complete fine-tuning datasets for more performance gain; 2) Since larger layer groups (e.g., 16) achieve a

Table 1: Performance comparison on LLaMA2-7B of domain-specific (upper part) and multi-domain mix-up FFT settings (lower part). Each part has three blocks containing 1) common LoRA and its continued variant (green), 2) FFT serving as an upper bound and its continued variant (blue), and 3) our MFT (red) with the other two vanilla masking baselines. The pre-trained model performance is shown on top, serving as a lower bound in our evaluation.

| Method | | Math | | Coding | | Instruction Following | |
|---|---|---|---|---|---|---|---|
| | | GSM8K | Math | HumanEval | HumanEval+ | IF-Eval | Alpaca-Eval |
| | Pre-Trained Model | 15.2 | 2.5 | 25.8 | 22.4 | 34.3 | 0.5 |
| Specific Domain | Best LoRA | $34.8_{\pm0.64}$ | $4.7_{\pm0.26}$ | $28.7_{\pm0.56}$ | $23.8_{\pm0.48}$ | $35.7_{\pm0.68}$ | $1.2_{\pm0.16}$ |
| | Continued LoRA | $33.6_{\pm0.72}$ | $4.5_{\pm0.23}$ | $28.2_{\pm0.49}$ | $23.2_{\pm0.43}$ | $32.5_{\pm0.84}$ | $1.1_{\pm0.12}$ |
| | **Best FFT** | $46.8_{\pm0.52}$ | $6.6_{\pm0.21}$ | $29.4_{\pm0.36}$ | $25.1_{\pm0.34}$ | $41.2_{\pm0.62}$ | $1.9_{\pm0.18}$ |
| | Continued FFT | $45.0_{\pm0.58}$ ↓1.8 | $5.5_{\pm0.23}$ ↓1.1 | $27.9_{\pm0.42}$ ↓1.5 | $23.6_{\pm0.38}$ ↓1.5 | $37.8_{\pm0.68}$ ↓3.4 | $2.0_{\pm0.21}$ ↑0.1 |
| | Random Mask w/ Best FFT | $38.4_{\pm0.68}$ | $4.9_{\pm0.25}$ | $28.0_{\pm0.51}$ | $24.0_{\pm0.47}$ | $39.4_{\pm0.99}$ | $1.4_{\pm0.21}$ |
| | L1 Mask w/ Best FFT | 42.6 | 5.7 | 28.9 | 24.5 | 40.9 | 1.5 |
| | MFT w/ Best FFT (**Ours**) | $\mathbf{47.3}_{\pm0.49}$ ↑0.5 | $\mathbf{7.4}_{\pm0.24}$ ↑0.8 | $\mathbf{31.8}_{\pm0.31}$ ↑2.4 | $\mathbf{27.9}_{\pm0.25}$ ↑2.8 | $\mathbf{44.1}_{\pm0.72}$ ↑2.9 | $\mathbf{3.0}_{\pm0.22}$ ↑1.1 |
| Mixed Domain | Best LoRA | $40.8_{\pm0.60}$ | $6.1_{\pm0.28}$ | $22.6_{\pm0.55}$ | $18.3_{\pm0.44}$ | $37.3_{\pm0.81}$ | $0.7_{\pm0.11}$ |
| | Continued LoRA | $31.9_{\pm0.54}$ | $4.0_{\pm0.21}$ | $20.1_{\pm0.42}$ | $17.1_{\pm0.36}$ | $31.5_{\pm0.92}$ | $0.8_{\pm0.09}$ |
| | **Best FFT** | $45.5_{\pm0.48}$ | $8.1_{\pm0.26}$ | $29.7_{\pm0.43}$ | $26.7_{\pm0.38}$ | $43.6_{\pm0.68}$ | $1.0_{\pm0.15}$ |
| | Continued FFT | $44.1_{\pm0.52}$ ↓1.4 | $7.5_{\pm0.27}$ ↓0.6 | $21.1_{\pm0.58}$ ↓8.6 | $18.1_{\pm0.43}$ ↓8.6 | $38.6_{\pm0.74}$ ↓5.0 | $1.2_{\pm0.16}$ ↑0.2 |
| | Random Mask w/ Best FFT | $40.0_{\pm0.63}$ | $6.5_{\pm0.32}$ | $23.7_{\pm0.57}$ | $19.1_{\pm0.49}$ | $30.0_{\pm0.95}$ | $0.9_{\pm0.14}$ |
| | L1 Mask w/ Best FFT | 43.3 | 7.7 | 25.7 | 22.3 | 32.8 | 1.2 |
| | MFT w/ Best FFT (**Ours**) | $\mathbf{45.9}_{\pm0.47}$ ↑0.4 | $\mathbf{8.4}_{\pm0.28}$ ↑0.3 | $\mathbf{31.5}_{\pm0.46}$ ↑1.8 | $\mathbf{27.3}_{\pm0.41}$ ↑0.6 | $\mathbf{46.1}_{\pm0.71}$ ↑2.5 | $\mathbf{1.9}_{\pm0.18}$ ↑0.9 |

Table 2: Performance comparison on LLaMA3.1-8B of domain-specific (upper part) and multi-domain mix-up FFT settings (lower part). Each part has three blocks containing 1) common LoRA and its continued variant (green), 2) FFT serving as an upper bound and its continued variant (blue), and 3) our MFT (red) with the other two vanilla masking baselines. The pre-trained model performance is shown on top, serving as a lower bound in our evaluation.

| Method | | Math | | Coding | | Instruction Following | |
|---|---|---|---|---|---|---|---|
| | | GSM8K | Math | HumanEval | HumanEval+ | IF-Eval | Alpaca-Eval |
| | Pre-Trained Model | 55.9 | 14.6 | 42.1 | 37.8 | 17.6 | 9.0 |
| Specific Domain | Best LoRA | $70.0_{\pm0.68}$ | $22.7_{\pm0.22}$ | $50.0_{\pm0.32}$ | $45.1_{\pm0.43}$ | $49.1_{\pm0.92}$ | $10.1_{\pm0.51}$ |
| | Continued LoRA | $67.1_{\pm0.75}$ | $19.7_{\pm0.19}$ | $46.7_{\pm0.46}$ | $44.5_{\pm0.38}$ | $44.3_{\pm0.88}$ | $8.9_{\pm0.29}$ |
| | **Best FFT** | $76.8_{\pm0.58}$ | $24.2_{\pm0.18}$ | $51.0_{\pm0.39}$ | $44.9_{\pm0.31}$ | $59.6_{\pm0.69}$ | $11.9_{\pm0.36}$ |
| | Continued FFT | $74.4_{\pm0.57}$ ↓2.4 | $23.9_{\pm0.17}$ ↓0.3 | $48.6_{\pm0.36}$ ↓2.4 | $41.9_{\pm0.30}$ ↓3.0 | $57.5_{\pm0.67}$ ↓2.1 | $11.3_{\pm0.34}$ ↓0.6 |
| | Random Mask w/ Best FFT | $74.9_{\pm0.77}$ | $22.3_{\pm0.23}$ | $48.2_{\pm0.48}$ | $42.7_{\pm0.41}$ | $58.8_{\pm0.96}$ | $11.3_{\pm0.48}$ |
| | L1 Mask w/ Best FFT | 75.1 | 22.7 | 49.1 | 44.1 | 59.8 | 11.5 |
| | MFT w/ Best FFT (**Ours**) | $\mathbf{77.3}_{\pm0.54}$ ↑0.5 | $\mathbf{24.8}_{\pm0.21}$ ↑0.6 | $\mathbf{53.5}_{\pm0.41}$ ↑2.5 | $\mathbf{46.8}_{\pm0.40}$ ↑1.9 | $\mathbf{65.6}_{\pm0.70}$ ↑6.0 | $\mathbf{13.7}_{\pm0.32}$ ↑1.8 |
| Multi Domain | Best LoRA | $72.8_{\pm0.82}$ | $24.5_{\pm0.29}$ | $63.1_{\pm0.82}$ | $55.5_{\pm0.65}$ | $38.2_{\pm1.18}$ | $11.0_{\pm0.64}$ |
| | Continued LoRA | $70.4_{\pm0.76}$ | $24.1_{\pm0.27}$ | $59.1_{\pm0.75}$ | $54.3_{\pm0.59}$ | $32.1_{\pm1.05}$ | $5.3_{\pm0.32}$ |
| | **Best FFT** | $72.7_{\pm0.56}$ | $24.4_{\pm0.24}$ | $64.4_{\pm0.59}$ | $60.2_{\pm0.48}$ | $60.0_{\pm0.73}$ | $11.8_{\pm0.48}$ |
| | Continued FFT | $71.7_{\pm0.54}$ ↓1.0 | $24.2_{\pm0.20}$ ↓0.2 | $62.6_{\pm0.53}$ ↓1.8 | $58.3_{\pm0.44}$ ↓1.9 | $59.7_{\pm0.80}$ ↓0.3 | $8.8_{\pm0.38}$ ↓3.0 |
| | Random Mask w/ Best FFT | $72.5_{\pm0.79}$ | $23.2_{\pm0.28}$ | $58.5_{\pm0.73}$ | $55.5_{\pm0.62}$ | $57.7_{\pm1.22}$ | $10.0_{\pm0.51}$ |
| | L1 Mask w/ Best FFT | 72.5 | 24.1 | 63.4 | 59.1 | 58.9 | 10.7 |
| | MFT w/ Best FFT (**Ours**) | $\mathbf{73.6}_{\pm0.60}$ ↑0.9 | $\mathbf{25.2}_{\pm0.23}$ ↑0.8 | $\mathbf{66.3}_{\pm0.51}$ ↑1.9 | $\mathbf{61.4}_{\pm0.50}$ ↑1.2 | $\mathbf{62.3}_{\pm0.69}$ ↑2.3 | $\mathbf{12.2}_{\pm0.34}$ ↑0.4 |

significant performance drop, we omit their results in Fig. 3 for a better illustration and leave them in the supplementary.

**Comparisons.** Tab. 1 and Tab. 2 show our main comparisons on LLaMA2-7B and LLaMA3.1-8B backbones. We start from a pre-trained model, followed by two FFT settings using the fine-tuning dataset of specific and mixed domains for each backbone. For each setting, we make sufficient FFT and report its best and continued versions (blue block), and we do the same for LoRA as well (green block). We also include random and L1 masks as two vanilla masking baselines (white block). MFT (red block) starts from the best FFT and achieves further performance gains. All three masking methods are based on the best FFT as a strong starting point. Among them, two critical baselines are the best FFT and continued FFT, where the former one serves as a competitive baseline and the latter one as a sanity check of potential performance drop caused by overfitting, and shows MFT effectiveness. We train our MFT only using domain-specific datasets for both FFT scenarios (domain-specific and mixed-up). Besides, we compare the training cost in Fig. 4 from the GPU memory, the number of used tokens, and the overall training time aspects on A100 GPUs. We use the coding domain as an example, and more comparisons are in the supplementary. Compared with the most important baseline, Best FFT, MFT involves very limited overhead for both token usage and training time. Compared with the Continued FFT, which also requires additional training overhead at least the same or even more, based on the Best FFT, MFT needs less training cost while achieving performance improvements. Since MFT fixes all model parameters and only conducts local mask learning, it always has an advantage in memory cost.

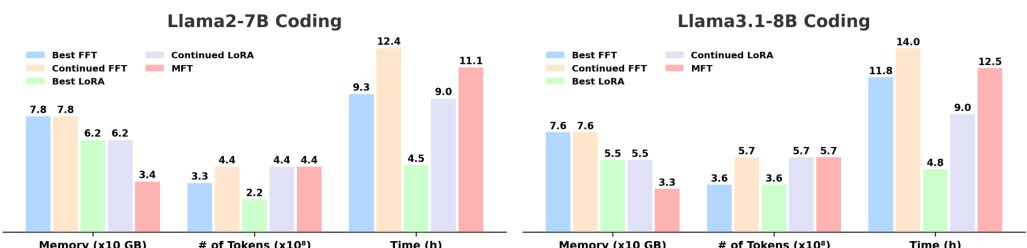

Figure 4: Training cost comparisons of GPU memory, used token number, and training time aspects. We use the coding domain as an example to compare MFT with other FFT and LoRA baselines.

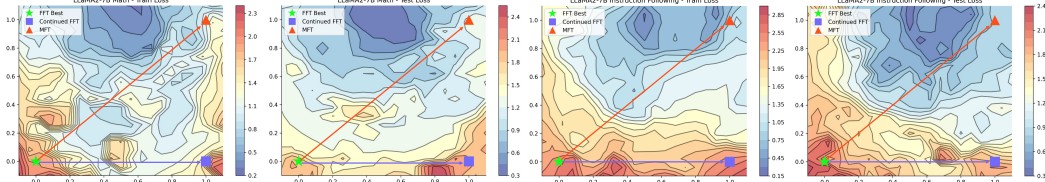

Figure 5: The loss landscape visualization on math and instruction following domains using LLaMA2-7B. Such visualization indicates that the proposed MFT further refines the Best FFT model to a better optimization and generalization status.

Accordingly, we conclude: 1) Since the Continued FFT causes a performance drop and the Best FFT outperforms other comparison methods, it is reasonable to set it as our strong baseline. 2) MFT achieves promising and consistent improvements across different backbones, domains, and FFT configurations. It accordingly upgrades the current routine as a new LLM fine-tuning protocol by learning and applying a mask on a well-trained model. Besides, the other two masking baselines generally damage the performance, especially compared with consistent improvements by MFT, which indicates that MFT is not a trivial process and is valuable to explore.

### 3.3 ANALYSES

**Theoretical Analysis.** We follow the PAC-Bayes theory about the generalization upper bound (McAllester, 1998), which also has been widely explored for neural networks (Arora et al., 2018), to provide theoretical intuitions from the information theory perspective. We focus on the performance gain from the model after the Best FFT to the model after MFT. Since our MFT uses exactly the same training objective and dataset as FFT, the analysis is under a fair comparison scenario to purely compare the model generalization upper bound.

Given $n$ as the size of the training set, $\delta$ as the degree of confidence, for any hypothesis $h$, which represents model here, we have the PAC-Bayes upper bound given by

$$L(h) \leq L_S(h) + \Phi(C(h)), \quad \Phi(u) = \sqrt{\frac{u + \ln \frac{1}{\delta}}{2(n-1)}}. \tag{6}$$

$L(h)$ and $L_S(h)$ represent training and test loss, respectively. $\Phi(C(h))$ is an additional complexity term to describe the model code length, where $C(\cdot)$ represents the encoding process. Then, we have

$$L(h_{\text{FFT}}) \leq L_S(h_{\text{FFT}}) + \Phi(C_{\text{FFT}}) \Rightarrow U(h_{\text{FFT}}) = L_S(h_{\text{FFT}}) + \Phi(C_{\text{FFT}}), \tag{7}$$

$$L(h_{\text{MFT}}) \leq L_S(h_{\text{MFT}}) + \Phi(C_{\text{MFT}}) \Rightarrow U(h_{\text{MFT}}) = L_S(h_{\text{MFT}}) + \Phi(C_{\text{MFT}}), \tag{8}$$

where $U(*)$ means the loss upper bound of $h$. If we make a difference on two sides, we have

$$U(h_{\text{MFT}}) - U(h_{\text{FFT}}) = \left[ L_S(h_{\text{MFT}}) - L_S(h_{\text{FFT}}) \right] + \left[ \Phi(C_{\text{MFT}}) - \Phi(C_{\text{FFT}}) \right], \tag{9}$$

revised as

$$U(h_{\text{MFT}}) - U(h_{\text{FFT}}) = \Delta_{\text{train}} + \Delta_{\text{complexity}}. \tag{10}$$

We numerically show $\Delta_{\text{train}} + \Delta_{\text{complexity}} < 0$. Due to limited space, details are left in the Appendix.

**Mask Fine-Tuning Loss Landscape.** Loss surface visualization is commonly used to analyze the model optimization process, sharing the insight of model status from a loss perspective. Thus, we visualize the loss surface of MFT to illustrate its training dynamics. Fig. 5 shows the surface of LLaMA2-7B on math and instruction following domains. The Best FFT, continued FFT, and MFT

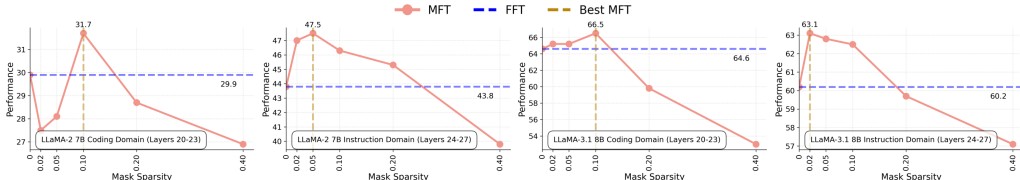

Figure 6: Masking ratio ablation visualizations. We use coding and instruction domains on LLaMA2 and LLaMA3.1. We observe the original 10% ratio works well on coding but not the optimal one on instruction, which indicates the masking ratio matters and more MFT potential.

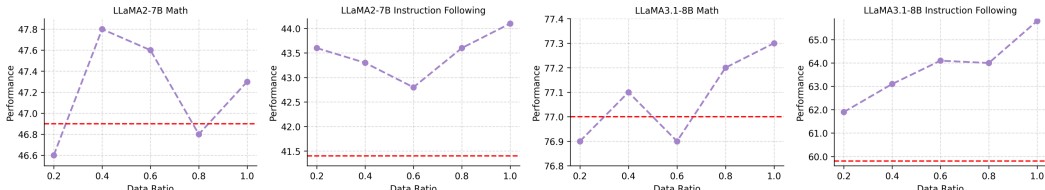

Figure 7: Data ratio ablation visualizations. We use math and instruction domains on LLaMA2 and LLaMA3.1. Compared with Best FFT (red dashed line), we observe MFT (purple) always improves the model using the full dataset, but may still obtain a promising performance gain using less data.

are represented by the green star, the purple rectangle, and the red triangle, respectively. We find MFT always optimizes the model to a better status compared with the Best FFT start point and its continued version for both training and test scenarios. We supplement more theoretical analyses of the loss landscape in the Appendix.

**Masking Ratio Ablation.** We ablate the masking ratio for local MFT. We consider 10% as the masking ratio for all settings (Tab. 1 and Tab. 2), which may not be an optimal option. Herein, we set a series of masking ratios for ablation. Specifically, we use all three domains on two backbones and set the masking ratio from 0.02 to 0.4. Results of coding and instruction are shown in Fig. 6, and the math results are in the supplementary. We find our original ratio of 10% works well on certain coding cases. However, it is not optimal for the instruction domain, where a smaller ratio works better than 10%. Such observation indicates that the masking ratio of MFT does matter to the final performance.

**Data Ratio Ablation.** We ablate data ratios for MFT. Fig. 7 shows the performance trends of math and instruction following domains on LLaMA2-7B and LLaMA3.1-8B (others are in the supplementary). We find MFT (purple) with a full dataset always leads to model improvement compared with the Best FFT (red dashed line), however, less data may also perform competitively in a more efficient way.

**Global Masking Fine-Tuning.** Different from the local MFT above, we initially explore MFT in a global way. Instead of pre-defining the sparse ratio, we set a threshold (-0.035 here) for the weight score $c_l$ in global MFT, which keeps the weights with scores larger than the threshold or otherwise. We expect it to encourage the MFT to automatically detect the appropriate sparse ratio. We use the math domain

Table 3: Initial exploration of global MFT on the math domain using LLaMA2-7B and LLaMA3.1-8B backbones. We observe better performance on LLaMA2-7B, especially for GSM8K, but a performance drop on LLaMA3.1-8B.

| Method | LLaMA2-7B | | LLaMA3.1-8B | |
|---|---|---|---|---|
| | GSM8K | Math | GSM8K | Math |
| Pre-Trained Model | 15.2 | 2.5 | 55.9 | 14.6 |
| Best FFT | 46.9 | 6.7 | 77.0 | 24.3 |
| Continued FFT | 45.2 ↓ 1.7 | 5.7 ↓ 1.0 | 74.6 ↓ 2.4 | 24.0 ↓ 0.3 |
| Global MFT w/ Best FFT (**Ours**) | 49.0 ↑ 2.1 | 7.1 ↑ 0.4 | 74.1 ↓ 2.9 | 21.8 ↓ 2.5 |

on LLaMA2-7B and LLaMA3.1-8B for the global MFT. The results in Tab. 3 show that the global MFT performs better than the local one using LLaMA2 on GSM8K, but causes a performance drop using LLaMA3.1-8B. Such results indicate the global mask has more potential for further improvement, but may require more systematic investigation, and we leave it to our future work.

# 4 RELATED WORK

This work explores optimizing a well-trained large language model (LLM) by learning a binary mask and proposes mask fine-tuning (MFT), upgrading the current LLM fine-tuning protocol. We summarize relevant literature of our work from (1) LLM pre-training and fine-tuning and (2) sparse network perspectives, emphasizing their correlations and differences.

## 4.1 Large Language Models Pre-training & Fine-tuning

LLM pre-training leads to significant performance gain for general language model capacity for both understanding (Devlin et al., 2019; Raffel et al., 2020; Liu, 2019; He et al., 2021) and generation (Radford et al., 2019; Brown et al., 2020; Achiam et al., 2023; Grattafiori et al., 2024; Touvron et al., 2023). Massive language corpora with large-scale pre-training enable them to be strong language backbones with basic common knowledge. Following pre-training, downstream fine-tuning based on small but high-quality target datasets further enhances pre-trained models with specific capacities of target domains (Hendrycks et al., 2021; Yu et al., 2024; Cobbe et al., 2021; Chen et al., 2021; Austin et al., 2021; Zhou et al., 2023; Wang et al., 2023; Nguyen, 2023). Different fine-tuning optimization strategies deliver various LLM properties, such as supervised fine-tuning (SFT) follows a pre-training objective to involve extra domain knowledge into LLM (Taori et al., 2023; Chiang et al., 2023; Wang et al., 2022b), and policy-based fine-tuning methods better align the human preference with LLM (Schulman et al., 2017; Rafailov et al., 2024; Ivison et al., 2024; Ouyang et al., 2022; Dai et al., 2024). We propose a mask fine-tuning (MFT) as a post-fine-tuning strategy to further refine the fine-tuned LLM, especially for SFT in this study. Such a strategy can be flexibly deployed after any fine-tuned model to propose a new protocol to improve LLMs.

## 4.2 Sparse Networks

Model sparsity in neural networks is close to neural network pruning (Liu et al., 2019; Molchanov et al., 2017; Wang et al., 2022a) for model compression (Han et al., 2016b; Iandola, 2016) and acceleration (Han et al., 2016a; Wang et al., 2021; Wang & Fu, 2023; Ma et al., 2023; Fang et al., 2023). Our work shares some overlaps with network pruning, since both aim to find a model partition with certain sparsity, however, pruning tries to remove a large part of the model for efficiency, while ours focuses on removing parameters irrelevant or even harmful to model capacities, which may only take a small model partition. More importantly, pruning methods are always accompanied by a performance loss of the trained model, while ours focuses on its further enhancement. In addition, the sparsity concept is also involved in other machine learning fields such as network optimization (Srivastava et al., 2014; Srinivas et al., 2017; Baldi & Sadowski, 2013), model architecture design (Shazeer et al., 2017; Zoph, 2017; Elsken et al., 2019), lottery ticket hypothesis (Frankle & Carbin, 2019; Zhou et al., 2019; You et al., 2020; Bai et al., 2022b), neural network mechanism (Wortsman et al., 2020; Ramanujan et al., 2020; Bai et al., 2022a), etc. In our study, we introduce the sparsity concept to explore if all parameters are beneficial for a well-trained LLM and further improve the model by learning and applying a mask on it, proposing a new way for LLM fine-tuning.

## 5 Discussion and Conclusion

**Intuition.** MFT is highly relevant to a series of well-established research topics like sparse network training, network pruning, etc. However, inspired by their techniques, we approach the sparsity concept in a novel way, from typical model efficiency to a general capability perspective by removing negative components for better performance. In other words, sparse models typically aim to compress the model while maintaining performance (seen as *subtraction*); however, we use sparsity as a tool for model enhancement (seen as *addition*). We expect that such counter-thinking of sparse networks could inspire more general explorations in the future to enrich their research potential.

**Limitation.** Due to a vast number of pre-trained models, domains, and evaluation benchmarks in the community, this work only considers representative ones to deliver a complete exploration pipeline. In addition, we only involve pure language models in this work, and extending it into the multi-modal area will further support our statement and conclusion. Through our exploration, we observe interesting phenomena, including the sensitivity trend of MFT (Fig. 3), MFT showing promising results with very limited optimization steps (Fig. 1), initial try of global MFT not consistently working well (Fig. 3), etc. Detailed investigation of these points potentially further strengthens the MFT.

**Conclusion.** We challenge the necessity of large language models (LLMs) structural integrity to study if a well-trained LLM can be further improved by removing certain model parameters. We propose a mask fine-tuning (MFT) to explore it based on the model with sufficient full fine-tuning (FFT). It consistently makes further improvements, generally improves LLM capability, and naturally upgrades the LLM fine-tuning pipeline. Comprehensive experiments with detailed ablations support our conclusion with exploration intuitions, covering different pre-trained LLM backbones, downstream domains, and fine-tuning configurations. Meanwhile, MFT also extends the functionality of model sparsity from compression to a general capability scenario, which inspires more future work.

ETHICS STATEMENT

In our paper, we strictly follow the ICLR ethical research standards and laws. To the best of our knowledge, our work abides by the General Ethical Principles.

REPRODUCIBILITY STATEMENT

We adhere to ICLR reproducibility standards and ensure the reproducibility of our work. All datasets we employed are publicly available. We will provide the code to reviewers and area chairs using an anonymous link if needed in the discussion period.

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

# APPENDIX

## A    IMPLEMENTATION DETAILS

We conduct our experiments on 8*A100 GPUs. To make full fine-tuning (FFT) sufficient, we set 4 as the maximum epoch number for FFT, where we observe over-fitting with a performance drop. We fine-tune MFT with a total of 2 epochs, which is addtional training epochs starting from the best FFT checkpoint (within total of 4 epochs). We set batch size as 8, accumulation steps as 16, and weight decay as 0.0 for all settings. We use 2e-5 and 2e-6 learning rates for LLaMA2-7B and LLaMA3.1-8B with linear schedule. Warm-up learning rate starts from 0 and linearly increases, and the warm-up ratio is set as 0.03 of the whole training. We summarize details of the training and test dataset used for our experiments in Tab. 4.

Table 4: The information of training and test datasets used in our experiments.

| Domain | Train / Test | Dataset Name | # of Samples |
|---|---|---|---|
| Math | Train | Tulu 3 Persona MATH Lambert et al. (2025) | 149,960 |
| | | Tulu 3 Persona GSM Lambert et al. (2025) | 49,980 |
| | | Tulu 3 Persona Algebra Lambert et al. (2025) | 20,000 |
| | | MetaMathQA Yu et al. (2024) | 395,000 |
| | | NuminaMath-TIR LI et al. (2024) | 64,312 |
| | Test | GSM8K Cobbe et al. (2021) | 1,320 |
| | | MATH Lewkowycz et al. (2022) | 5,003 |
| Coding | Train | Evol CodeAlpaca Luo et al. (2024) | 107,276 |
| | | Code-Alpaca Chaudhary (2023) | 20,022 |
| | | Tulu 3 Persona Python Lambert et al. (2025) | 34,999 |
| | Test | HumanEval Chen et al. (2021) | 164 |
| | | HumanEval+ Liu et al. (2023) | 164 |
| Instruction Following | Train | OpenAssistant Guanaco Köpf et al. (2023) | 7,132 |
| | | Tulu 3 Persona IF Lambert et al. (2025) | 29,980 |
| | | Open-Orca Mukherjee et al. (2023) | 30,000 |
| | Test | IFEval Zhou et al. (2023) | 100 |
| | | AlpacaEval 2.0 Li et al. (2023) | 805 |

# B  MORE EVALUATIONS

Tab. 5 and Tab. 6 extend the performance tables in our main draft by adding cross-domain evaluations for our MFT, FFT, and other comparison methods, under domain-specific FFT settings. The best performance for each setting is highlighted in bold.

Table 5: Cross-domain evaluations on LLaMA2-7B under domain-specific FFT setting.

| Method | Math | | Coding | | Instruction Following | |
|---|---|---|---|---|---|---|
| | GSM8K | Math | HumanEval | HumanEval+ | IF-Eval | Alpaca-Eval |
| **Pre-trained Model** | 15.2 | 2.5 | 25.8 | 22.4 | 34.3 | 0.5 |
| **Best FFT** [Math] | 46.9 | 6.7 | 17.7 | 15.9 | 29.3 | 1.5 |
| **Best FFT** [Coding] | 14.6 | 2.8 | 29.3 | 25.0 | 8.3 | 1.4 |
| **Best FFT** [IF] | 25.0 | 2.4 | 16.5 | 13.4 | 41.4 | 1.7 |
| Continued FFT [Math] | 45.2 | 5.7 | 19.5 | 17.1 | 33.2 | 1.5 |
| Continued FFT [Coding] | 11.1 | 2.5 | 28.0 | 23.8 | 13.1 | 2.7 |
| Continued FFT [IF] | 23.1 | 2.1 | 13.1 | 12.8 | 37.4 | 2.0 |
| Random Mask w/ Best FFT [Math] | 38.4 | 4.9 | 14.8 | 13.1 | 25.5 | 1.0 |
| Random Mask w/ Best FFT [Coding] | 12.4 | 2.5 | 28.0 | 24.0 | 7.5 | 1.1 |
| Random Mask w/ Best FFT [IF] | 23.0 | 2.3 | 14.4 | 12.2 | 39.4 | 1.5 |
| L1 Mask w/ Best FFT [Math] | 42.6 | 5.7 | 16.4 | 14.6 | 28.3 | 1.1 |
| L1 Mask w/ Best FFT [Coding] | 12.8 | 2.6 | 28.9 | 24.5 | 7.8 | 1.0 |
| L1 Mask w/ Best FFT [IF] | 23.9 | 2.4 | 14.9 | 12.7 | 40.9 | 1.5 |
| **[Ours]** Best FFT + Mask [Math] | **47.3** | **7.6** | 16.5 | 14.6 | 30.1 | 0.5 |
| **[Ours]** Best FFT + Mask [Coding] | 13.8 | 2.8 | **31.7** | **28.0** | 8.5 | 2.1 |
| **[Ours]** Best FFT + Mask [IF] | 23.5 | 2.8 | 15.2 | 11.6 | **44.1** | **2.9** |

Table 6: Cross-domain evaluations on LLaMA3.1-8B under domain-specific FFT setting.

| Method | Math | | Coding | | Instruction Following | |
|---|---|---|---|---|---|---|
| | GSM8K | Math | HumanEval | HumanEval+ | IF-Eval | Alpaca-Eval |
| **Pre-trained Model** | 55.9 | 14.6 | 42.1 | 37.8 | 17.6 | 9.0 |
| **Best FFT** [Math] | 77.0 | 24.3 | 35.4 | 29.9 | 31.5 | 0.6 |
| **Best FFT** [Coding] | 60.7 | 13.8 | 51.2 | 45.1 | 5.3 | 0.9 |
| **Best FFT** [IF] | 63.0 | 10.1 | 50.0 | 44.5 | 59.8 | 12.0 |
| Continued FFT [Math] | 74.6 | 24.0 | 33.5 | 31.1 | 32.2 | 0.7 |
| Continued FFT [Coding] | 59.6 | 13.1 | 48.8 | 42.1 | 4.8 | 1.4 |
| Continued FFT [IF] | 63.6 | 8.7 | 45.7 | 43.5 | 57.2 | 11.4 |
| Random Mask w/ Best FFT [Math] | 74.9 | 22.3 | 34.3 | 28.5 | 30.0 | 0.4 |
| Random Mask w/ Best FFT [Coding] | 56.2 | 12.7 | 48.2 | 42.7 | 4.7 | 0.7 |
| Random Mask w/ Best FFT [IF] | 61.3 | 9.2 | 45.3 | 41.7 | 58.8 | 11.5 |
| L1 Mask w/ Best FFT [Math] | 75.1 | 22.7 | 34.4 | 30.6 | 31.1 | 0.6 |
| L1 Mask w/ Best FFT [Coding] | 57.3 | 12.9 | 49.1 | 44.1 | 4.8 | 0.8 |
| L1 Mask w/ Best FFT [IF] | 62.3 | 9.3 | 46.1 | 42.4 | 59.8 | 11.3 |
| **[Ours]** Best FFT + Mask [Math] | **77.3** | **24.9** | 36.0 | 32.2 | 31.1 | 0.5 |
| **[Ours]** Best FFT + Mask [Coding] | 58.6 | 13.6 | **53.7** | **47.0** | 4.9 | 0.0 |
| **[Ours]** Best FFT + Mask [Instruction] | 63.2 | 9.6 | 44.5 | 40.9 | **65.8** | **13.8** |

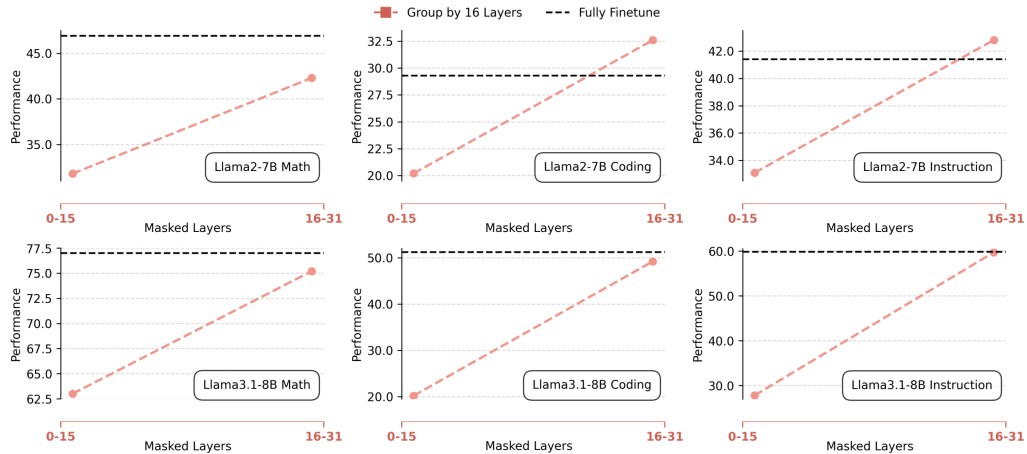

Figure 8: The local MFT ablation results of the larger layer-wise group (16-layer).

## C  MORE ABLATIONS AND ANALYSES

**Larger Layer-wise Group Ablation.** The results of larger layer-wise group (16-layers) ablation for local MFT is shown in Fig. 8.

**Fine-grained Evaluation Details.** We provide fine-grained evaluation details for FFT and MFT on three domains using LLaMA2-7B in Tab. 7 (domain-specific FFT) and Tab. 8 (domain mixed-up FFT). They show the epoch number of Best FFT, Continued FFT, Best MFT, and Complete MFT with corresponding token usage information. The number of Continued FFT is consistent at 4, and those for Best FFT are less than 4. Based on Best FFT, we further conduct 2 epochs for MFT. Therefore, the number of Complete MFT is always equal to the number of Best FFT plus 2, but Best MFT may be less than it. For example, for the math domain in Tab. 7, Continual FFT is 4 epochs, while FFT achieves the best results at 2.4 epochs as Best FFT. Starting from 2.4 epochs, we make MFT with an additional 2 epochs, thus, Complete MFT is 2.4 + 2 = 4.4 epochs, while Best MFT is 4 epochs, which means MFT uses 4 - 2.4 = 1.6 epochs to achieve the best result based on Best FFT. For each setting, the corresponding used token number is provided in brackets.

Table 7: Training epochs and corresponding number of used tokens (in brackets) on LLaMA2-7B under domain-specific FFT setting.

| Domain | Best FFT | Continued FFT | Best MFT | Complete MFT |
|---|---|---|---|---|
| Math | 2.4 epochs (5.90e8) | 4.0 epochs (9.83e8) | 4.0 epochs (9.83e8) | 4.4 epochs (1.08e9) |
| Coding | 2.7 epochs (3.32e8) | 4.0 epochs (4.92e8) | 3.6 epochs (4.43e8) | 4.7 epochs (5.78e8) |
| Instruction Following | 2.4 epochs (5.51e8) | 4.0 epochs (9.18e8) | 3.2 epochs (7.35e8) | 4.4 epochs (1.01e9) |

Table 8: Training epochs and corresponding number of used tokens (in brackets) on LLaMA2-7B under mixed-up FFT setting.

| Domain | Best FFT | Continued FFT | Best MFT | Complete MFT |
|---|---|---|---|---|
| Math | 2.0 epochs (2.67e9) | 4.0 epochs (5.34e9) | 3.7 epochs (3.09e9) | 4.0 epochs (3.16e9) |
| Coding | 2.0 epochs (2.67e9) | 4.0 epochs (5.34e9) | 3.3 epochs (2.83e9) | 4.0 epochs (2.92e9) |
| Instruction Following | 2.0 epochs (2.67e9) | 4.0 epochs (5.34e9) | 4.0 epochs (3.13e9) | 4.0 epochs (3.13e9) |

**More Details of Theoretical Analysis.** For our case, to theoretically support that our MFT has better optimization potential than FFT, we need to verify that $\Delta_{\text{train}} + \Delta_{\text{complexity}} < 0$ in Eq. 10.

For the first term, we first supplement a set of exemplar training loss statistics in Tab. 9. It shows the mean value of the training loss after the training is stable, which indicates that MFT can further reduce the training loss compared with the model after FFT (the Best FFT), on which it is based.

Table 9: Exemplar training loss statistics of LLaMA2-7B models after training is stable.

| LLaMA2-7B | Math | Coding | Instruction |
|---|---|---|---|
| FFT loss | 0.101 | 0.098 | 0.125 |
| MFT loss | 0.085 | 0.054 | 0.060 |

Then, we can ensure the first term $\Delta_{\text{train}} < 0$.

For the second term, we need to compare the encoded model complexity. Given $d$ as the total number of weights, $z$ as the weights being masked out (as zero), and $b$ as the bit width. Since we are using weight-level masking granularity, we have

$$C(h_{\text{FFT}}) \approx bd, \quad C(h_{\text{MFT}}) \approx b(1-p)d + \log_2 \binom{d}{z}, \tag{11}$$

where $p = z/d$ represents the sparse ratio in our case. The first term of $C_{\text{MFT}}$ is the encoding cost of non-zero weights, and the second is for the index cost of zero weights. Further, we have $\log_2 \binom{d}{z} \approx dH(p)$, where

$$H(p) = -p \log_2 p - (1-p) \log_2(1-p). \tag{12}$$

Then, the second term can be described as

$$\Delta_{\text{complexity}} = C(h_{\text{MFT}}) - C(h_{\text{FFT}}) \approx d[H(p) - bp]. \tag{13}$$

For practice, we always use $b = 8$ or $16$. We use $10\%$ sparse ratio for our MFT, and it is applied on a 4-layer group, then it results in a global sparse ratio of $2.7\% \sim 3\%$ for two backbones. To consider the strict case, we use $b = 8$ and a ratio of $2.7\%$ for a calculation case shown below.

$$H(p) - bp = H(0.027) - 8 \cdot 0.027 \approx -0.037 < 0. \tag{14}$$

Then, we have the second term $\Delta_{\text{complexity}} < 0$.

Combining both the first and second terms, we have

$$U(h_{\text{MFT}}) - U(h_{\text{FFT}}) = \Delta_{\text{train}} + \Delta_{\text{complexity}} < 0. \tag{15}$$

Finally, we conclude that $U(h_{\text{MFT}}) < U(h_{\text{FFT}})$, which means the MFT can further reduce the test loss upper bound compared with FFT and theoretically support our proposed method.

**More Training Cost Comparisons.** We supplement training cost comparisons of GPU memory, used token number, and training time for math and instruction following domains in Fig. 9.

**More Mask Ratio Ablation.** The mask ratio ablation on the math domain using LLaMA2-7B and LLaMA3.1-8B is shown in Fig.10.

**More Data Ratio Ablation.** The MFT data ratio ablation on coding domain using LLaMA2-7B and LLaMA3.1-8B is shown in Fig. 11.

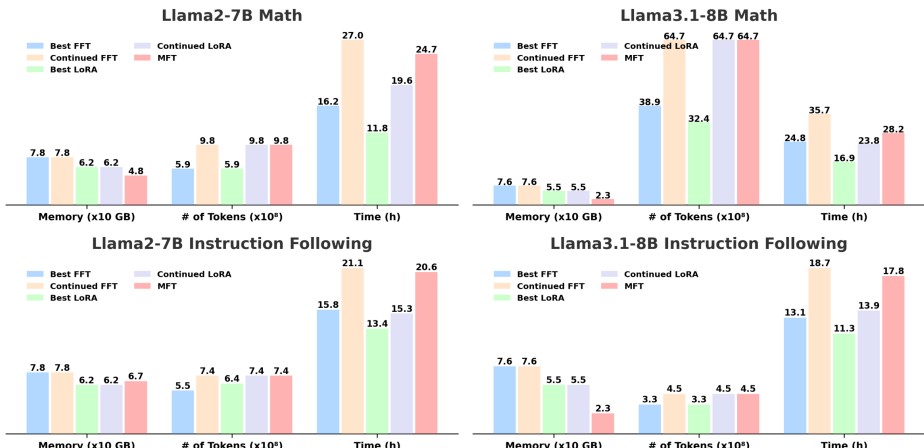

Figure 9: Training cost comparisons of GPU memory, used token number, and training time on math and instruction following domain using LLaMA2-7B and LLaMA3.1-8B.

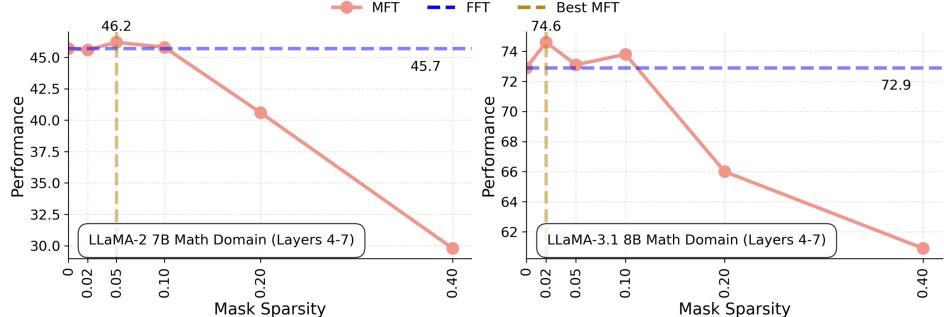

Figure 10: Masking ratio ablation study on the math domain using LLaMA2-7B and LLaMA3.1-8B.

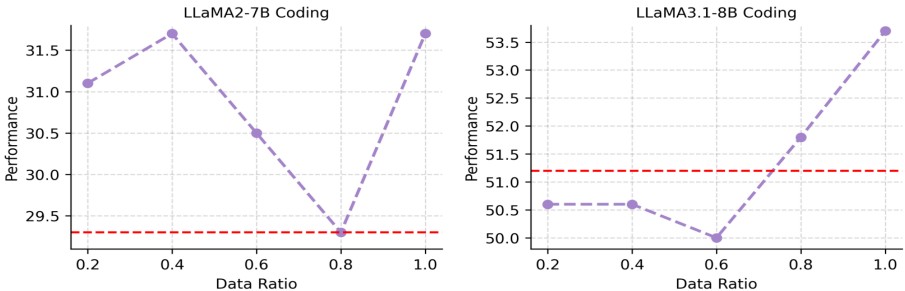

Figure 11: Data ratio ablation study on coding domain using LLaMA2-7B and LLaMA3.1-8B.

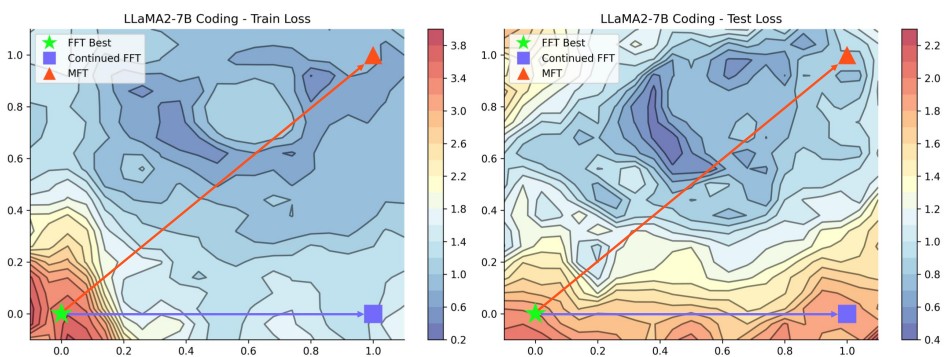

Figure 12: The loss landscape visualization on the coding domain using LLaMA2-7B.

**More Details of Loss Landscape Visualization.** Loss landscape visualization on the coding domain using LLaMA2-7B is shown in Fig. 12.

For the loss surface analysis, we provide the empirical analysis as loss landscape visualization in Fig. 5 and Fig. 12. Here, we supplement a theoretical view to further support it.

We assume the model after FFT (Best FFT) has Hessian as

$$H_{\text{FFT}} = \nabla^2 L(\theta_{\text{FFT}}) \succeq 0. \tag{16}$$

Since MFT learns a mask and applies it on the model after FFT, which means the MFT model is a projection $P$ of the FFT model, where the corresponding Hessian can be given by

$$H_{\text{MFT}} = PH_{\text{FFT}}P. \tag{17}$$

Since for $H_{\text{FFT}} \succeq 0$ and its any projection ($H_{\text{MFT}}$ for here), we have

$$\lambda_{\max}(H_{\text{MFT}}) \leq \lambda_{\max}(H_{\text{FFT}}), \quad \text{Tr}(H_{\text{MFT}}) \leq \text{Tr}(H_{\text{FFT}}). \tag{18}$$

As $\lambda_{\max}$ and $\text{Tr}(\cdot)$ represent the model sharpness and mean curvature, lower values of them indicate the model has a flatter status, which means the MFT model has better generalization property than the FFT model. Such analysis serves as a theoretical support, and it also corresponds to the empirical results shown in the landscape visualizations (the MFT model is visually flatter than the FFT one).

## D MASK VISULIZATION

In Fig. 13 and Fig. 14, we visualize the head-level mask sparsity patterns learned on LLaMA2-7B of the coding domain under 10% sparsity, using six different random seeds. The patterns across all six seeds are remarkably consistent. This indicates that MFT performs fine-grained, weight-level subnetwork selection that subtly reweights specific attention directions. The convergence of zero-value locations across seeds further demonstrates that, for the same model and domain, MFT learns a structured, stable, and reproducible subnetwork rather than random noise. These results align with our design intuition that MFT identifies domain-relevant structural preferences with high consistency.

## E THE USE OF LARGE LANGUAGE MODELS (LLMS)

We only use the LLM to fix grammar errors.

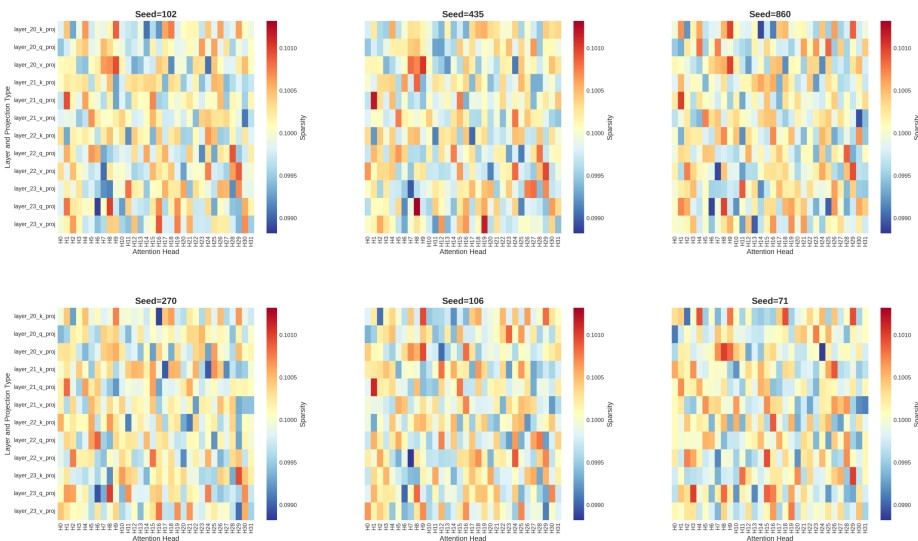

Figure 13: Head-level mask sparsity distributions learned on LLaMA2-7B of the coding domain under 10% sparsity, using six different random seeds. Each heatmap shows the average sparsity values of Q/K/V projections across attention heads for layers 20–23.

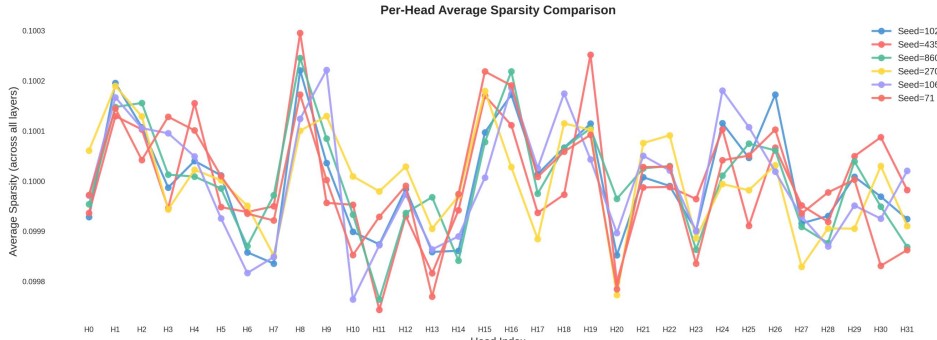

Figure 14: The visualization of average sparsity across layers of each head for six different seeds under the same 10% sparsity configuration on LLaMA2-7B of coding domain (layers 20–23). All six curves exhibit highly similar fluctuation patterns across the 32 heads, showing that MFT consistently learns the same attention directions across seeds.

