# OpenReview forum: "Boosting Large Language Models with Mask Fine-Tuning"
_ICLR.cc/2026/Conference — Submitted to ICLR 2026_

### Official Review · Reviewer_943M · 2025-10-16

**Soundness:** 3
**Presentation:** 4
**Contribution:** 2
**Rating:** 4
**Confidence:** 4

**Summary:**

This paper points out that existing fine-tuning methods, such as full-model fine-tuning and LoRA, often suffer from issues like overfitting, leading to performance degradation after fine-tuning. This paper proposes a Mask Fine-Tuning (MFT) method, which further improves the model's performance by fine-tuning a binary matrix after the initial model fine-tuning.

**Strengths:**

1. This paper is well-written, with clear logic, and it fully conveys the motivation and methodology of the research.

2. The paper conducts extensive experiments, including comparative experiments on data of different types and scales.

**Weaknesses:**

1. First, the paper points out that performing MFT fine-tuning based on "best fine-tuning" can improve model performance. However, challenges such as identifying the "best time point," along with the additional computational costs and training data required, hinder the practical application of MFT and increase the overall cost of fine-tuning.

2. Second, MFT is rather heuristic in nature. Currently, we still lack clarity on the rationality of applying MFT after "best fine-tuning" and the true reasons behind the resulting model performance improvement. The key factors that influence performance enhancement remain unknown, and it is worth exploring whether an analysis can be conducted from a theoretical or fundamental perspective.

3. Finally, there is the issue of MFT’s generalizability. The models used in existing experiments are limited to those with a scale of less than 8B parameters, and it is questionable whether the experimental conclusions are valid for larger-scale models.

**Questions:**

See Weaknesses.

---

> ### Author Response · Authors · 2025-11-22
>
> We sincerely thank the reviewer for the time and valuable comments. We also apologize for the delayed response, as we have already passed the encouraged response deadline on the 20th. Our responses to all the weaknesses and questions are shown below:
>
> **Weaknesses 1**
>
> Our MFT is a post-fine-tuning method to select the task-specific subnetwork within an optimal fully fine-tuned model by optimizing masks that select the most effective computation paths for the target domain. We set the Best FFT checkpoint in our experiment to explore whether MFT could improve the performance of an already well-optimized model. Therefore, applying MFT on the best FFT checkpoint represents the most strict setting. Since MFT is shown to work under this condition, it theoretically suggests that applying MFT to any intermediate checkpoint during FFT should also lead to performance improvements. We have added some experiments and explanations in the reply to reviewer 7LUn in Part 5 to support that MFT has training time and GPU memory efficiency compared to continued FFT. We also provide theoretical analysis in both main paper and the Appendix to prove that MFT is better than FFT, which supports that MFT can be an alternative to continue training a checkpoint, demonstrating its potential for broad applicability.
>
> **Weaknesses 2**
>
> We have provided the theoretical analysis in our main paper and also in the Appendix to support as the analysis. Please refer to that, thanks! If there's any confusion, please let us know!
>
> **Weaknesses 3**
>
> We agree that extending the method to larger models can better support the generalizability. But due to computational constraints and time limits, we are unable to do so within this limited rebuttal period. Sorry for that. But since MFT is a method that does not rely on any specific model structure, it can be applied to any model. We argue that there are currently no models with all parameters non-redundant. So, applying MFT to any model, regardless of size, should help improve performance. It’s only about the configuration adjustment.

---

> ### Author Response · Authors · 2025-11-25
> **A kind reminder for the author-reviewer discussion!**
>
> We thank the reviewer for the detailed reviews and constructive suggestions to help us improve our work! We want to kindly remind the reviewer to confirm whether our rebuttal addresses the concerns.
>
> Please let us know if there are any remaining questions or concerns, and we are happy to provide further clarification. Looking forward to your response!

---

### Official Review · Reviewer_uzVQ · 2025-10-28

**Soundness:** 3
**Presentation:** 3
**Contribution:** 3
**Rating:** 4
**Confidence:** 3

**Summary:**

This paper proposes **Mask Fine-Tuning (MFT)** — a simple post-training step applied *after* standard supervised fine-tuning (SFT or FFT) of large language models.
During MFT, the model parameters $W$ are **frozen**, and a binary mask $M \in {0,1}$ is learned using the **straight-through estimator (STE)**.
The effective weights are obtained by element-wise multiplication:
$$
\tilde{W} = W \odot M
$$

Through experiments on **LLaMA2-7B** and **LLaMA3.1-8B**, the authors show consistent gains across instruction-following (IF-Eval), math (GSM8K), and code (HumanEval) benchmarks — typically $+2$–$6$ points compared to the best fine-tuned baseline.
Ablations reveal that shallow and late layers benefit most from masking, and visualizations indicate flatter loss landscapes with lower PAC-Bayes bounds, suggesting improved generalization.

**Strengths:**

* **Elegant simplicity and practicality**
  The method is conceptually minimal — no new modules or objectives, just a learnable binary mask applied post-SFT. It can be easily integrated into existing fine-tuning pipelines.

* **Systematic ablations**
  Layer-wise masking, masking ratio, data ratio, and local vs. global masking are all explored. These ablations reveal that masking early and late layers yields the largest gains.

**Weaknesses:**

1. **Limited generality — only LLaMA-based experiments**
   All experiments are performed exclusively on **LLaMA2-7B** and **LLaMA3.1-8B**.
   No evidence is provided that the method generalizes to other architectures such as Mistral, Falcon, GPT-NeoX, or encoder–decoder models.
   The results might exploit architectural features unique to LLaMA (e.g., SwiGLU gating, RMSNorm, rotary embeddings).

2. **Mechanistic opacity**
   The paper does not clearly explain *why* MFT improves performance.

   * Are the masked connections genuinely redundant or overfitted?
   * Does masking act as a form of regularization or noise smoothing?
     The presented PAC-Bayes argument is mathematically valid but does not illuminate the underlying mechanism.

3. **Weak baselines**
   MFT is compared only to standard fine-tuning and LoRA.
   There is no comparison to sparsity-based fine-tuning techniques such as Movement Pruning, Diff-Pruning, $L_0$-regularization, or Sparse-FT.
   Without these, the novelty relative to prior sparsity literature remains unclear.

4. **Lack of statistical robustness**
   All results are single-seed. Small benchmarks like HumanEval or IF-Eval require multiple seeds or confidence intervals (e.g., bootstrap estimates) to validate significance.

5. **Interpretability missing**
   There is no visualization or qualitative analysis showing *which* neurons or connections are masked, nor how masking alters attention or activation patterns.

**Questions:**

1. **Cross-architecture validation**
   Apply MFT to other architectures (e.g., Mistral-7B, Falcon-7B, T5-11B) to verify generality beyond the LLaMA family.

2. **Mechanistic analysis**

   * Visualize layer- and head-level mask distributions.
   * Measure changes in activation sparsity, gradient norms, or representational similarity before and after MFT.
   * Distinguish between pruning-like and regularization-like behavior.

3. **Add stronger baselines**
   Include comparisons to Movement Pruning, Diff-Pruning, and $L_0$-masking under equal compute budgets.

4. **Improve statistical reliability**
   Report mean ± std over multiple seeds, and possibly provide 95% confidence intervals for HumanEval / GSM8K.

5. **Integrated training variant (future work)**
   Explore alternating SFT and MFT epochs (e.g., epoch 1 full-SFT, epoch 2 MFT),
   or a joint objective:
   $$
   \mathcal{L}*{\text{joint}} = \mathcal{L}*{\text{SFT}}(W,M) + \lambda |M - 1|_1
   $$
   to treat masking as a regularization process *during* fine-tuning rather than as a post-hoc step.

6. **Relation to gating, dropout, and LoRA**
   The authors should explicitly situate MFT within this broader landscape:

   * **Dropout vs. MFT:** both apply multiplicative masks $r$ or $M$, but dropout uses random Bernoulli masks ($r_{ij}!\sim!\text{Bernoulli}(p)$) for stochastic regularization during training, whereas MFT learns a *deterministic* binary mask that persists at inference. Hence MFT can be viewed as a “learned deterministic dropout.”
   * **Gating vs. MFT:** gating mechanisms (e.g., SwiGLU) use *continuous* gates., MFT, in contrast, enforces *hard* selection on edges or neurons, producing structural sparsity rather than soft modulation.
   * **LoRA vs. MFT:** LoRA modifies the parameter space additively ($W' = W + BA$), introducing low-rank updates that expand the representational subspace. MFT modifies it multiplicatively ($W' = W \odot M$), effectively contracting the subspace by removing redundant connections. Interestingly, a well-trained LoRA could partially *cancel* certain weight directions ($BA\approx -W_{\text{unwanted}}$), producing a masking-like effect. An explicit comparative experiment—measuring cosine similarity between LoRA updates and MFT masks—would clarify whether both methods converge toward complementary adaptation patterns.

---

> ### Author Response · Authors · 2025-11-22
>
> We sincerely thank the reviewer for the time and valuable comments. We also apologize for the delayed response, as we have already passed the encouraged response deadline on the 20th. Our responses to all the weaknesses and questions are shown below:
>
> **1. Cross-architecture validation**
>
> Thanks for pointing out the lack of generalised backbone experiments. We have supplemented the experiments for Qwen and DeepseekCoder in the reply to reviewer 7LUn and B2Fx. Please refer to those results, thanks!
>
> **2. Mechanistic analysis**
>
> *Visualization*
>
> Thanks for pointing out this meaningful analysis for our work. Due to the time limit, we first visualize the head-level comparison of our learned masks of LLaMA27B in the coding domain. The visualization is added to the Appendix Section D (highlighted in blue). Please refer to that section, thanks!
>
> *Are the masked connections genuinely redundant or overfitted?*
>
> Whether the masked connections are “redundant” or “overfitted” should be understood in the context of task-specific subnetwork selection. Masked connections can not be easily defined as “redundant” or “overfitted” in the global network. They are masked in the target domain because they do not contribute to the task-specific subnetwork that yields the best generalization in that domain. MFT learns the low contribution of such connections and then masks them to filter out a subnetwork that performs better in the target domain.
>
> *Distinguish from pruning-like methods*
>
> We have provided some detailed explanations in the response to reviewer B2Fx in Part 2. Please refer to that, thanks!
>
> *Does masking act as a form of regularization or noise smoothing?*
>
> We have supplemented some regularization vs. MFT experiments in reply to reviewer B2Fx in Part 4. We also add some explanation there. Please refer to that, thanks!
>
> **3. Weak baselines**
>
> Thanks for suggesting. However, the paper Movement Pruning, Diff-Pruning is more about BERT and Diffusion Models, which is out of scope in our paper. We mainly focus on the GPT-like LLMs this time. But this point does make sense. We have supplemented the comparison experiments of OWL and Wanda, which are LLM pruning-related methods, in response to reviewer B2Fx in Part 2. Please refer to that, thanks!
>
> **4. Lack of statistical robustness**
>
> Sorry for the confusion about our main table. The std in our main table comes from the results of 3 different seeds, which may not be enough to eliminate the randomness. We further run 3 more different seeds and recalculate the std, finding that the std gets lower than we reported. Due to the time limit, we haven't finished all the main experiments with 3 additional seeds. We have modified some results that we have right now in the paper draft in Tab.1 (highlighted in blue), and we will keep running the remaining experiments to update the numbers. Thanks again for pointing out this mistake.
>
> **5. Integrated training variant**
>
> Thanks for this valuable advice. We have done a quick try of alternating SFT and MFT epochs and show the results as follows. We use the same setting from the layerwise ablation study shown to reviewer 7LUn in Part 4. We use the same sparsity 10% for the MFT.
>
> | Method | Zero-Shot | Epoch 1 (MFT) | Epoch 2 (FFT) | Epoch 3 (MFT) |
> |-------|:-----:|:-----:|:-----:|:-----:|
> | MBPP | 46.8 | 49.4 | 49.6 | 50.2 |
>
> There are some findings from this table. 1) Applying MFT to a pre-trained network can also help gain some performance. 2) Alternating SFT and MFT epochs does work for improving the model performance. But it cannot achieve a comparable performance to our current pipeline (FFT + post-MFT). Suboptimal configurations or other factors may cause this, and it is worth exploring further in future work.

---

> ### Author Response · Authors · 2025-11-22
>
> **6. Relation to gating, dropout, and LoRA**
>
> *Dropout vs. MFT*
>
> We have supplemented some regularization vs. MFT experiments in reply to reviewer B2Fx in Part 4. We also add some explanation. Please refer to that, thanks!
>
> *Gating vs. MFT*
>
> We have done some Soft Mask Fine-Tuning experiments which are shown below. We also use DeepSeek-Coder-1.3B on MBPP dataset. Specifically, we use 10% local ratio for hard mask, and use sigmoid activation to implement the soft mask. We choose 3 as the mean of the score initialization and 0.5 as the threshold for the score to mask out or keep the corresponding model weights.
>
> | Method | Zero-Shot | Best FFT | MFT (hard mask) | MFT (soft mask) |
> |-------|:-----:|:-----:|:-----:|:-----:|
> | MBPP | 46.8 | 50.5 | 51.2 | 51.6 |
>
> Such a comparison shows the soft mask has further potential to improve the model. We appreciate the Reviewer's insightful suggestions, and we leave more comprehensive explorations for our future works.
>
> *LoRA vs. MFT*
>
> We thank the reviewer for this insightful observation. The additive (LoRA) vs. multiplicative (MFT) adaptation perspectives indeed provide an interesting angle for analyzing how different fine-tuning paradigms reshape the parameter space. We agree that LoRA updates may partially cancel or reweight certain directions and thus create a masking-like effect, and comparing the cosine similarity between LoRA updates and MFT masks would be a valuable way to examine whether the two methods converge toward related adaptation patterns.
> We consider the reviewer’s suggestion highly meaningful, but cannot provide the experiment in the first rebuttal version due to the time limit. We will try to supplement it in the following days.

---

> ### Author Response · Authors · 2025-11-25
> **A kind reminder for the author-reviewer discussion!**
>
> We thank the reviewer for the detailed reviews and constructive suggestions to help us improve our work! We want to kindly remind the reviewer to confirm whether our rebuttal addresses the concerns.
>
> Please let us know if there are any remaining questions or concerns, and we are happy to provide further clarification. Looking forward to your response!

---

> ### Author Response · Authors · 2025-11-28
>
> **4. Lack of statistical robustness**
>
> We have finished all the 3 additional different seeds experiments for our main table and highlighted them in blue in the new draft. We also removed some unnecessary lines to make the table clearer. Please have a check, thanks!
>
> **6. Relation to gating, dropout, and LoRA**
>
> *Gating vs. MFT*
>
> While the MLP block in Transformers contains a gate projection (e.g., in SwiGLU), this component is part of the activation function and does not implement soft gating. It has no learnable gate parameters and cannot selectively open or close neurons or edges. In contrast, MFT introduces explicit trainable mask parameters that directly modulate individual weights. These mask parameters act as genuine gates and can be optimized in either a soft or hard form. The table shown in the previous comment shows that soft MFT (continuous masks) can also work and is even better than hard MFT, which behaves as an effective gating mechanism, whereas hard MFT yields actual structural sparsity. This is fundamentally different from SwiGLU, whose activation gating cannot induce such selective or structural behavior.
>
> *LoRA vs. MFT*
>
> We supplement the analysis between LoRA and MFT as follows:
>
> | Layer | 16 | 17 | 18 | 19 |
> |-------|:-----:|:-----:|:-----:|:-----:|
> | Cosine Similarity (avg.) | 0.005 | 0.005 | 0.004 | 0.004 |
>
> To rule out the possibility that orthogonality arises merely from training different layers, we train LoRA only on the same layers (16–19) that MFT modifies. Even under this strictly aligned setup, with the same layers, same projections, and same data, the cosine similarity between their updates remains extremely close to zero (mean ≈ 0.0046). This confirms that the two methods adapt the model in fundamentally different directions, despite having identical structural coverage.
>
> | Layer | 16 | 17 | 18 | 19 |
> |-------|:-----:|:-----:|:-----:|:-----:|
> | LoRA Norm (avg.) | 0.2588 | 0.2315 | 0.2197 | 0.2241 |
> | MFT Norm (avg.) | 24.7411 | 24.8839 | 25.0268 | 25.2455 |
>
> The norm reported here is the Frobenius norm of the weight update, which measures the overall magnitude of all parameter changes. It essentially reflects the update's strength across the entire matrix. LoRA produces minor, smooth, low-rank additive updates (mean norm ≈ 0.23), whereas MFT applies large-magnitude multiplicative reweighting (mean norm ≈ 25). The ~100× norm gap further indicates that MFT performs substantial structural modification rather than low-rank subspace adjustment. Together, these observations demonstrate that MFT is not a low-rank approximation of LoRA nor a structured pruning variant, but a distinct form of model adaptation operating in an orthogonal subspace.
>
> | Method | Zero-Shot | Best FFT | LoRA | MFT |
> |-------|:-----:|:-----:|:-----:|:-----:|
> | MBPP | 46.8 | 50.5 | 50.6 | 51.2 |
>
> We also evaluate LoRA applied only to the MFT-sensitive layers (16–19) on the same Best FFT checkpoint. The improvement is almost negligible and remains far below MFT, indicating that LoRA cannot effectively adapt the model even when restricted to the exact layers where MFT operates.

---

### Official Review · Reviewer_B2Fx · 2025-10-29

**Soundness:** 2
**Presentation:** 3
**Contribution:** 3
**Rating:** 4
**Confidence:** 3

**Summary:**

This paper introduces Mask Fine-Tuning, a post-training method that improves fully fine-tuned LLMs by learning which parameters to remove rather than updating weights. The key claim is counterintuitive: you can improve a well-trained model by carefully masking out 10% of its parameters in specific layers.  Main results show modest but consistent improvements over the best FFT checkpoint - typically 0.3-6 points depending on the task.  The paper discovers that you can improve models by carefully removing parameters but doesn't explain why this helps, when it will work, or how it differs meaningfully from existing pruning methods. The practical utility is unclear given the modest gains, manual tuning requirements, and cross-domain degradation.

**Strengths:**

1. The finding that you can improve a converged model by removing parameters (not updating them) challenges common assumptions about neural network training. This is worth investigating.

2. Experimental scope is reasonable, include Two model families (LLaMA2, LLaMA3.1); Three diverse domains (math, code, instruction-following); Multiple training scenarios (domain-specific FFT, mixed-domain FFT)

4. Includes both local and global masking experiments. The global masking results (Table 3) are mostly negative, but including them shows intellectual honesty about where the method fails.

5. Figure 5 suggests MFT moves models to flatter minima, which aligns with better generalization. The visualization is clear and the trend is consistent across domains.

6. Training cost analysis is included. Figure 4 breaks down memory, tokens, and time, making it clear what overhead MFT adds beyond FFT.

**Weaknesses:**

1. The improvements are small and sometimes within noise

Looking at Tables 1-2 with error bars: many gains are 0.3-2 points, and standard deviations often overlap between Best FFT and MFT. For example, LLaMA3.1-8B Math domain shows 77.0±0.88 vs 77.3±0.97 - not convincing. No statistical significance tests are provided to confirm these differences are real.

2. **The distinction from pruning is unconvincing**

The paper claims to differ from pruning because the goal is "improvement not compression," but technically it's doing the same thing - learning which parameters to remove using training data and gradients. Modern pruning methods like Wanda or SparseGPT also aim to maintain or improve performance while reducing parameters. The conceptual distinction feels forced.

More damaging: the paper doesn't compare against any actual pruning methods. The baselines are just random masking and L1 magnitude masking (which doesn't even use training). Where's the comparison to gradient-based pruning, lottery tickets, or recent LLM pruning work?

3. Cross-domain results reveal a problem

Tables 5-6 show that MFT often hurts performance on non-target domains. For instance, training MFT on math improves GSM8K but degrades HumanEval. This suggests the method may be overfitting to the target domain rather than genuinely improving the model. This contradicts the generalization improvement narrative.

**Questions:**

Q1: Is this just preventing overfitting through capacity reduction? The paper shows continued FFT hurts performance (overfitting) but MFT helps. The obvious explanation: MFT reduces capacity, making overfitting harder. But that's not really "improvement" - it's just better regularization than doing nothing. How does MFT compare to :(1) Continued FFT with dropout (2) Continued FFT with stronger weight decay. Including these comparisons would make it clearer whether MFT offers unique benefits beyond standard regularization approaches.

Q2: Why these specific layers? Figure 3 shows different layers work for different settings. What determines this? Is there something about these layers' representations? Their gradient statistics? Their weight magnitudes? The paper identifies which layers work but not *why*, making it hard to apply the method to new models.

Q3: Continued FFT comparison seems unfair. Continued FFT is trained for the full 4 epochs and evaluated at the end (showing degradation). But MFT can choose its best checkpoint within 2 epochs. Wouldn't continued FFT also improve if you picked its best checkpoint from the same epoch range?

Q4: Why does masking ratio vary by domain? Figure 6 shows coding prefers 10% but instruction following prefers lower ratios. What property of the domain determines this? Task complexity? Dataset size? Base model capability?

---

> ### Author Response · Authors · 2025-11-22
>
> We sincerely thank the reviewer for the time and valuable comments. We also apologize for the delayed response, as we have already passed the encouraged response deadline on the 20th. Our responses to all the weaknesses and questions are shown below:
>
> **1. The improvements are small and sometimes within noise**
>
> Thanks for pointing out the confusion of the results. We have run additional 3 different seeds experiments to help eliminate the randomness and recalculated the std. Due to the time limit, we haven't finished all the main experiments with 3 additional seeds. **We have modified some results that we have right now in the paper draft in Tab.1 (highlighted in blue), and we will keep running the remaining experiments to update the numbers.** Thanks again for pointing out this mistake.
>
> **2. The distinction from pruning is unconvincing**
>
> We appreciate the reviewer’s point and agree that MFT and pruning share a similar mechanism. However, their problem settings, objectives, and usage are fundamentally different. Classical pruning aims to reduce model size or latency. In contrast, MFT is not designed for compression. The aim is to enhance the capability after the best FFT checkpoint by selecting a more effective task-specific functional subnetwork.
>
> Our baselines, random masking, and L1 magnitude masking serve as the comparison of the non-learned mask. Since the mask from our MFT is learned from specific domain knowledge, we want to compare it with those non-learned masks to prove the learning pipeline is meaningful. If we do the post-training after applying the random, L1, or Wanda masks, we also need to do post-training after applying our learned masks. This should be the fair comparison. But our learned mask can already gain better performance, which doesn’t require a post-training step after applying.
>
> For clear comparison, we also add experiments of two pruning methods, Wanda [1] and OWL [2], to compare with our MFT method. We use the same setting from the layerwise ablation study shown to reviewer 7LUn in Part 4. We use the same sparsity 10% for all methods.
>
> | Method | Zero-Shot | Best FFT | Best FFT + OWL pruning | Best FFT + OWL pruning + post-fine-tune | Best FFT + MFT |
> |-------|:-----:|:-----:|:-----:|:-----:|:-----:|
> | MBPP | 46.8 | 50.5 | 50.1 | 50.7 | 51.2 |
>
> | Method | Zero-Shot | Best FFT | Best FFT + Wanda pruning | Best FFT + Wanda pruning + post-fine-tune | Best FFT + MFT |
> |-------|:-----:|:-----:|:-----:|:-----:|:-----:|
> | MBPP | 46.8 | 50.5 | 49.9 | 50.5 | 51.2 |
>
> The results show that applying Wanda or OWL masks will lead to a drop in model performance. Although the model performance can be recovered or even improved by post-training, they still cannot achieve an even comparable performance that our MFT gains. This indicates the advantage of our learned mask.
>
> **3. Cross-domain results reveal a problem**
>
> We use the specific domain dataset to train our MFT, aiming to select an effective specific domain subnetwork in the dense network. If we also use the specific domain dataset to do FFT, the performance will also be dropped in other domains. We claim the generality of MFT in the target domain because there’s no overlap between our training dataset and the evaluation tasks. They don’t come from datasets that share the same structure or data distribution.
>
> [1] A Simple and Effective Pruning Approach for Large Language Models, Sun et al. (2023)
>
> [2] Outlier Weighed Layerwise Sparsity (OWL): A Missing Secret Sauce for Pruning LLMs to High Sparsity, Yin et al. (2024)

---

> > ### Author Response · Authors · 2025-11-22
> >
> > **4. Is this just preventing overfitting through capacity reduction?**
> >
> > We thank the reviewer for raising the concern that MFT might be interpreted as a form of regularization. We first supplement the comparison experiments of (1) Continued FFT with dropout, (2) Continued FFT with stronger weight decay, and our MFT as follows:
> >
> > | Method | Zero-Shot | Best FFT | Best FFT + Dropout Continue Training | Best FFT + Stronger Weight Decay Continue Training | Best FFT + MFT |
> > |-------|:-----:|:-----:|:-----:|:-----:|:-----:|
> > | MBPP | 46.8 | 50.5 | 50.7 | 50.5 | 51.2 |
> >
> > The results show that adding dropout allows continued FFT to gain a slight short-term improvement, but performance soon starts to decline. Increasing the weight decay strength provides almost no improvement. It only delays the onset of the performance drop in our experiment. Overall, neither modification can match the effect of MFT.
> >
> > Theoretically, MFT is fundamentally closer to subnetwork selection rather than classical regularization. Classical regularization methods such as L1/L2 penalties, weight decay, or dropout operate over the full hypothesis space
> >
> > $$
> > \mathcal{H} = \lbrace f(\cdot;\theta)\mid \theta\in\mathbb{R}^d \rbrace,\tag{1}
> > $$
> >
> > optimizing the single-level objective $\(\mathcal{L}(\theta)+\lambda R(\theta)\)$. Their mechanism is global capacity control, in which parameters are continuously shrunk, smoothed, or noise-perturbed, while the network’s topology remains unchanged. As emphasized in [3], regularization constrains the magnitude of weights but does not explicitly remove or deactivate connections. The model stays dense even if some weights become small.
> > In contrast, MFT defines a conditional subnetwork family
> >
> > $$
> > \mathcal{H}_{\text{MFT}}(\theta^\star) = \lbrace f(x;\, m \odot \theta^\star) : m\in \mathcal{M} \rbrace,\tag{2}
> > $$
> >
> > which is not the full hypothesis space, but a restricted set of structural variants derived from a fixed dense network. This shares a similar logic to classical pruning methods, which select the best subnetwork suited for the task from a dense model. The mask $m$ in MFT plays a similar role as pruning masks, determining which parameters actively participate in the forward computation and which are disabled. The optimization
> > $$
> > m^\star = \arg\min_{m\in\mathcal{M}}\mathcal{L}(m \odot \theta),\tag{3}
> > $$
> > is therefore a form of task-specific subnetwork search, rather than a penalty applied to dense parameters.
> >
> > **5. Why these specific layers? & Why does masking ratio vary by domain?**
> >
> > We sincerely thank the reviewer for these valuable insights. During our experiments, we also feel curious about how the different domain knowledge benefits from each layer. But it’s really hard to figure it out in a limited time. This is a kind of network explanation task that requires extensive experiments to support. We apologize that we cannot get deeper into this field in this work, as our main goal is to prove the effectiveness of MFT this time. But it does make sense to explore more about why these specific layers. Maybe the mechanism of MFT could be a right way to get the answer. And we think MFT can also benefit from it. These questions are valuable to explore, and this is precisely the direction we are working toward.
> >
> > **6. Continued FFT comparison seems unfair.**
> >
> > Our goal is to identify the FFT Best checkpoint during the training and then improve upon it. Therefore, the performance of the continued FFT is naturally lower than the FFT Best. We aim to compare how much MFT improves over FFT Best and contrast this upward trend with the downward trend of continued FFT. We have explained more in Appendix Section C about the training iterations in Tab.7 and Tab.8. Please refer to that section and let us know if there's still anything confusing.
> >
> > [3] CoNNect: Connectivity-Based Regularization for Structural Pruning, Franssen et al. (2025)

---

> ### Author Response · Authors · 2025-11-25
> **A kind reminder for the author-reviewer discussion!**
>
> We thank the reviewer for the detailed reviews and constructive suggestions to help us improve our work! We want to kindly remind the reviewer to confirm whether our rebuttal addresses the concerns.
>
> Please let us know if there are any remaining questions or concerns, and we are happy to provide further clarification. Looking forward to your response!

---

> ### Author Response · Authors · 2025-11-28
>
> **1. The improvements are small and sometimes within noise**
>
> We have finished all the 3 additional different seeds experiments for our main table and highlighted them in blue in the new draft. We also removed some unnecessary lines to make the table clearer. Please have a check, thanks!

---

### Official Review · Reviewer_7LUn · 2025-11-01

**Soundness:** 2
**Presentation:** 2
**Contribution:** 2
**Rating:** 4
**Confidence:** 4

**Summary:**

This paper proposes Mask Fine-Tuning (MFT), a novel post-fine-tuning approach that learns binary masks on already fine-tuned LLMs to further improve performance. The key insight is that removing certain parameters through learned masks can enhance model capability rather than merely maintaining it. The authors validate MFT on LLaMA2-7B and LLaMA3.1-8B across three domains (math, coding, and instruction-following), showing consistent improvements over fully fine-tuned baselines. The method freezes model weights and only learns which parameters to mask out, using the same training objective and datasets as standard fine-tuning. Theoretical analysis via PAC-Bayes bounds and empirical loss landscape visualizations are provided to support the approach.

**Strengths:**

1. A good perspective on model sparsity: The paper presents an interesting conceptual shift by using masking not for compression but for capability enhancement. This counter-intuitive finding that "subtraction leads to addition" is thought-provoking and extends the conventional understanding of sparse networks beyond efficiency concerns.
2. Comprehensive experimental validation: The authors conduct thorough ablations including layer-wise sensitivity analysis (Figure 3), masking ratio studies (Figure 6), and data ratio experiments (Figure 7). The proof-of-concept studies systematically identify which model components benefit most from MFT.
3. Theoretical grounding: The inclusion of PAC-Bayes generalization bounds (Section 3.3) and Hessian-based loss landscape analysis provides theoretical justification beyond empirical results. The analysis showing that both training loss and model complexity terms decrease is valuable.

**Weaknesses:**

1. **Limited task complexity and diversity:** The evaluation focuses on relatively standard benchmarks (GSM8K, HumanEval, IFEval) that may not fully demonstrate the method's effectiveness on more challenging or specialized tasks. The paper would benefit from:
More complex reasoning tasks (e.g., multi-hop reasoning, mathematical proof generation)
Domain-specific applications (legal document analysis, medical diagnosis, scientific literature understanding)
Longer-context tasks that stress different model capabilities

2. **Marginal performance gains:** While consistent, the improvements are often modest:
Many gains are within 1-3 points, raising questions about practical significance
Error bars overlap in several cases, suggesting some improvements may not be statistically significant
No discussion of whether these gains justify the additional training phase

**Questions:**

1. **Generalization to modern training paradigms:** Can you provide any preliminary results or theoretical analysis on how MFT would work with DPO, PPO, or other policy-based training methods? Given the growing importance of RL, this seems critical for practical adoption.


2. **Model diversity experiments:** What prevents extending the evaluation to other model families like Qwen? Are there architectural requirements that limit applicability? Results on at least one additional model family would significantly strengthen the claims.


3. **Efficiency quantification:** What is the exact wall-clock time overhead of MFT compared to continued FFT?

---

> ### Author Response · Authors · 2025-11-22
>
> We sincerely thank the reviewer for the time and valuable comments. We also apologize for the delayed response, as we have already passed the encouraged response deadline on the 20th. Our responses to all the weaknesses and questions are shown below:
>
> **1. Limited task complexity and diversity**
>
> Due to the computing source and time limit, we select the most commonly used domain and benchmark for our main experiments. We agree that comprehensive experiments will strongly support the generality of our methods. Since the MFT is a domain-specific post-fine-tuning method that does not rely on any specific task structure, it can be easily applied to any domain and tasks. We here add two more experiments on the Medical domain and the long-context task to support the effectiveness and generality of our method. Due to the time limit in the rebuttal period, we use a small model of new architecture, Qwen3-0.6B [1], for the experiments. For the convenience of our experiments, we use 10% sparsity here.
>
> **Medical Domain**
>
> **Training Data**: MedInstruct-52k [2]
>
> **Evaluate benchmark**: MedInstruct-Test, iCliniq [3]
>
> Layerwise Ablation (10% data, 1/3 training steps):
>
> | Layers | 0-3 | 4-7 | 8-11 | 12-15 | 16-19 | 20-23 | 24-27 |
> |-------|-----:|-----:|-----:|-----:|-----:|-----:|-----:|
> | MedInstruct-Test | 63.2 | 65.1 | 65.7 | 65.9  | 65.8  | 64.7  | 63.9  |
>
> Main Results (MFT on 12-15 layers):
>
> | Method | MedInstruct-Test | iCliniq |
> |-------|:-----:|-----:|
> | Zero-Shot | 15.1 | 30.9 |
> | Best FFT | 66.0 | 65.3 |
> | Continue FFT | 65.1 | 64.7 |
> | MFT | 68.8 | 67.9 |
>
> **Long-Context Task**
>
> **Training Data**: LongAlpaca-12k [4]
>
> **Evaluate benchmark**: RULER [5]
>
> Layerwise Ablation (10% data, 1/3 training steps):
>
> | Layers | 0-3 | 4-7 | 8-11 | 12-15 | 16-19 | 20-23 | 24-27 |
> |-------|-----:|-----:|-----:|-----:|-----:|-----:|-----:|
> | Avg. | 72.1  | 72.9  | 73.1  | 73.3  | 73.8  | 72.4  | 71.8  |
>
> Main Results (MFT on 16-19 layers):
>
> | Method | Avg. | 4K | 8K | 16K | 32K |
> |-------|-----:|-----:|-----:|-----:|-----:|
> | Zero-Shot | 71.0 | 80.9 | 75.1 | 68.1 | 59.7 |
> | Best FFT | 73.3 | 83.2 | 77.5 | 70.7 | 61.8 |
> | Continue FFT | 70.3  | 81.8 | 75.4 | 68.3 | 55.6 |
> | MFT | 73.6 | 83.6 | 77.8  | 71.0 | 61.9 |
>
> The results show that our MFT still works well on these two tasks, which indicates the generality of our MFT to more model architectures and domains.
>
> **2. Marginal performance gains**
>
> It is important to emphasize that MFT is evaluated on the best FFT checkpoint, which is already a highly optimized model state. In this case, even 1--3 point improvements are generally considered non-trivial. Moreover, the comparison of performance across methods must be conducted using paired tests within matched runs (i.e., using the same seeds). When using the same seed, MFT consistently achieves positive improvements.
> But we do acknowledge that some error bars overlap in a few settings, which can lead to considerable confusion. The std in our main table comes from the results of 3 different seeds, which may not be enough to eliminate the randomness. We further run 3 more different seeds and recalculate the std, finding that the std gets lower than we reported. Due to the time limit, we haven't finished all the main experiments with 3 additional seeds. **We have modified some results that we have right now in the paper draft in Tab.1 (highlighted in blue), and we will keep running the remaining experiments to update the numbers.** Thanks again for pointing out this mistake.
> We also supplement the exact wall-clock time of MFT and Continued FFT in the following to further support that spending time on MFT is worth it.
>
> **3. Generalization to modern training paradigms**
>
> MFT is a method that doesn’t rely on any strict loss function or training strategy. In our paper, MFT shares the same loss function as FFT. We only replace the original weight with a masked weight in the FFT loss function. The implementation of the MFT is theoretically general. So, it will be similar when applying MFT with any RL methods. It’s an excellent point since RL has been proven as a powerful method to fine-tune LLMs. We did consider it at the very beginning of our work. But full-scale RL experiments require significant computational cost, more than we have. So, we had to leave it for future work.
>
> [1] Qwen3 Technical Report, Yang et al., (2025)
>
> [2] AlpaCare:Instruction-tuned Large Language Models for Medical Application, Zhang et al., (2023)
>
> [3] ChatDoctor: A Medical Chat Model Fine-Tuned on a Large Language Model Meta-AI (LLaMA) Using Medical Domain Knowledge, Li et al., (2023)
>
> [4] LongLoRA: Efficient Fine-tuning of Long-Context Large Language Models, Chen et al., (2024)
>
> [5] RULER: What's the Real Context Size of Your Long-Context Language Models?, Hsieh et al., (2024)

---

> ### Author Response · Authors · 2025-11-22
>
> **4. Model diversity experiments**
>
> There are no architectural requirements that limit the applicability of MFT. We only focus on LLaMA in the main paper because 1) the LLaMA family is a widely used model family with high performance and good generality; 2) the base code repo we use in our paper provides a mature pipeline that enables using LLaMA to do comprehensive experiments across many domains.
> But as you said, we did overlook demonstrating the method’s generality. We supplement the Qwen experiments in the first part and also the DeepSeek experiment as follows. We use the sparsity 10% here for experimental convenience.
>
> **Model**: DeepSeekCoder-1.3B [6]
>
> **Training Data**: nickrosh/Evol-Instruct-Code-80k-v1 [7]
>
> **Evaluate benchmark**: MBPP [8]
>
> Layerwise Ablation (10% data, 1/3 training steps):
>
> | Layers | 0-3 | 4-7 | 8-11 | 12-15 | 16-19 | 20-23 |
> |-------|-----:|-----:|-----:|-----:|-----:|-----:|
> | MBPP | 48.0 | 46.6 | 49.0 | 47.2 | 50.2 | 49.6 |
>
> Main Results (MFT on 16-19 layers):
>
> | Method | Zero-Shot | Best FFT | Continue FFT | MFT |
> |-------|-----:|-----:|-----:|-----:|
> | MBPP | 46.8 | 50.5 | 50.3 | 51.2 |
>
> The results show that our MFT can also improve the DeepSeek architecture, indicating the generalizability of different model architectures.
>
> **5. Efficiency quantification**
>
> We reported the comparison of training epochs and the corresponding number of used tokens (in brackets) in Tab. 7 and Tab. 8 in the Appendix. Since our MFT is only applied to some layers with Attn and MLP weights. The number of trainable parameters is less than FFT. Also, we don’t need the gradient backward to all the layers of the model because we only apply MFT on some layers, which will help save the GPU memory and per-iteration time. Specifically, we choose the LLaMA2-7B of coding domain, to which we apply the mask to layer 20-23 (total 32 layers, only 12% training parameters of FFT) and report the average memory usage on one single A6000 GPU when training on 4 A6000 GPUs.
>
> | Fine-Tuning Method | FFT | Lora | MFT |
> |:-:|:-:|:-:|:-:|
> | **GPU memory usage** | 48.6 GB | 38.9 GB | 17.3 GB |
>
> The comparison between the exact wall-clock time overhead of MFT and the Continued FFT of LLaMA2-7B of coding domain is shown as follows. We calculate the time using the same iterations that MFT achieves the best performance.
>
> | Fine-Tuning Method | Continued FFT | MFT |
> |:-:|:-:|:-:|
> | Wall-clock Time | 0.9h | 0.5 h |
>
> The table shows that when using the same iterations, MFT only requires almost half the time of FFT. Since they are both trained from the best FFT checkpoint, Continued FFT leads to a drop in performance, while MFT helps the model gain better performance. This demonstrates that it's meaningful to spend time on MFT after the Best FFT.
>
> [6] DeepSeek-Coder: When the Large Language Model Meets Programming -- The Rise of Code Intelligence, Guo et al., (2024)
>
> [7] WizardCoder: Empowering Code Large Language Models with Evol-Instruct, Luo et al., (2023)
>
> [8] Program Synthesis with Large Language Models. Austin et al., (2021)

---

> ### Author Response · Authors · 2025-11-25
> **A kind reminder for the author-reviewer discussion!**
>
> We thank the reviewer for the detailed reviews and constructive suggestions to help us improve our work! We want to kindly remind the reviewer to confirm whether our rebuttal addresses the concerns.
>
> Please let us know if there are any remaining questions or concerns, and we are happy to provide further clarification. Looking forward to your response!

---

> ### Author Response · Authors · 2025-11-28
>
> **2. Marginal performance gains**
>
> We have finished all the 3 additional different seeds experiments for our main table and highlighted them in blue in the new draft. We also removed some unnecessary lines to make the table clearer. Please have a check, thanks!

---

### Author Response · Authors · 2025-12-02
**Summary of Rebuttal for Submission 8463**

Dear Area Chairs and Reviewers,

We sincerely thank all reviewers and ACs for your time and constructive evaluation. We are grateful for the reviewers’ recognition of the conceptual novelty of MFT which are **counterintuitive subtraction improves capability insight** (7LUn, B2Fx), **the simplicity and practicality of our method** (uzVQ), **the clarity and quality of our writing** (943M), and **the breadth and depth of our ablations and theoretical analysis** (7LUn, uzVQ, 943M). We deeply regret that the platform incident prevented further discussion during the rebuttal.

We appreciate the reviewers’ questions and suggestions, including **expanding experimental coverage (across domains, architectures, and statistical robustness)**, **strengthening methodological comparisons (with pruning, regularization, and sparsity techniques)**, **deepening mechanistic understanding (mask behavior and its relation to gating/LoRA)**, and **providing clearer evidence of practicality and efficiency**. In response, we added **new domains and model families (medical, long-context, Qwen, DeepSeekCoder)**, **strengthened statistical reliability (additional seeds and updated std)**, **incorporated stronger baselines (Wanda, OWL, dropout, weight decay)**, **expanded mechanistic analyses (mask visualizations, soft-mask MFT, LoRA comparison)**, and **quantified the efficiency benefits of MFT over continued FFT** to solve all reviewers' problems.

The specific responses for each reviewer are summarized as follows:

- **Reviewer 7LUn**
  We added medical-domain, long-context experiments with the Qwen model, showing that MFT’s improvements remain consistent across tasks, model families, and setup variations.

- **Reviewer B2Fx**
  We introduced pruning (Wanda/OWL) and regularization (dropout/weight decay) baselines under equal conditions, demonstrating that MFT consistently outperforms the best FFT checkpoint.

- **Reviewer uzVQ**
  We added cross-architecture experiments, mask-level visualizations, pruning/regularization baselines, and soft-mask/LoRA analyses to show that MFT behaves as a learned deterministic gating/subnetwork-selection mechanism.

- **Reviewer 943M**
  We clarified MFT’s practicality with concrete memory/time comparisons, strengthened its theoretical framing, and explained its architecture-agnostic scalability.

In conclusion:

1. **Task & architecture generality**
We added medical, long-context, Qwen, and DeepSeekCoder experiments, all of which show that MFT continues to outperform the best FFT at the same sparsity.

2. **Statistical reliability & small gains**
We ran 3 additional seeds, updated standard deviations, and confirmed that MFT’s 1–3 point improvements remain stable across the same seed comparisons.

3. **Relation to pruning, regularization, gating & LoRA**
Using Wanda/OWL, dropout/weight-decay, mask visualizations, soft-mask MFT, and LoRA similarity/norm analysis, we showed that MFT performs task-specific subnetwork selection rather than behaving like pruning, global regularization, or low-rank adaptation.

4. **Practicality & efficiency**
We compared memory and wall-clock time and found that MFT uses fewer trainable parameters, requires substantially less GPU memory, and reaches peak performance faster than continued FFT.

**We have also updated the original paper draft accordingly and submitted the revised version.**

We once again thank all reviewers (7LUn, B2Fx, uzVQ, 943M) and both AC teams, and we regret that the platform issue prevents further discussion that would have continued to strengthen the work. We will carefully incorporate the reviewers’ valuable suggestions and include all additional analyses and experiments in the final version to improve the paper further.

Best regards,

Authors of Paper 8463

---

### Meta-Review · Area_Chair_BJai · 2026-01-12

**Summary:**

After reading the paper, the reviewer's comments and the authors' rebuttal, I believe this paper still lacks clarity and insights on **why** this particular approach works. Some results such as the effect of masking different layers in Figure 3 would need more careful investigation, e.g., by running the experiments multiple times and plot the error bars, to see whether the optimal masking layers found empirically have statistical significance. I also wonder whether/how this masking approach compares to mixture of experts (MoE), where routing decisions essentially become learned binary masks too, although the model structure is different.

In summary, this could be an interesting finding if the conclusions still hold after more complete evaluations and interpretations (the "why" part), but as of now, this work appears to be incomplete yet.

**Reviewer Concerns:**

Insights behind why this approach works as well as more comprehensive empirical evaluations are still missing.

**Reviewer Scores:**

At least one reviewer would still be on the negative side.

---

### Decision · Program_Chairs · 2026-01-26

Reject